# ParticleDA.jl v.1.0: A distributed particle filtering data assimilation package

Daniel Giles[1,2,*], Matthew M. Graham[2,*], Mosè Giordano[2], Tuomas Koskela[2], Alexandros Beskos[1,3], and Serge Guillas[1,2,3]

[1]Department of Statistical Sciences, University College London, London, UK
[2]Centre of Advanced Research Computing (ARC), University College London, London, UK
[3]The Alan Turing Institute, London, UK
[*]These authors contributed equally to this work.

**Correspondence:** Daniel Giles (d.giles@ucl.ac.uk)

**Abstract.** Digital twins of physical and human systems informed by real-time data, are becoming ubiquitous across weather forecasting, disaster preparedness, and urban planning, but researchers lack the tools to run these models effectively and efficiently, limiting progress. One of the current challenges is to assimilate observations in highly non-linear dynamical systems, as the practical need is often to detect abrupt changes. We have developed a software platform to improve the use of real-time data in non-linear system representations where non-Gaussianity limits the applicability of data assimilation algorithms such as the ensemble Kalman filter and variational methods. Particle filter based data assimilation algorithms have been implemented within a user-friendly open source software platform in Julia – ParticleDA.jl. To ensure the applicability of the developed platform in realistic scenarios, emphasis has been placed on numerical efficiency and scalability on high performance computing systems. Furthermore, the platform has been developed to be forward model agnostic, ensuring that it is applicable to a wide range of modelling settings, for instance unstructured and non-uniform meshes in the spatial domain or even state spaces that are not spatially organized. Applications to tsunami and numerical weather prediction demonstrate the computational benefits and ease of using the high-level Julia interface to the package to perform filtering in a variety of complex models.

## 1 Introduction

*Data assimilation* (DA) focuses on optimally combining observations with a dynamical model of a physical system to estimate how the system state evolves over time. The field of research has its origins within the *numerical weather prediction* (NWP) community, where DA techniques are applied iteratively to update current best estimates of the state of the atmosphere. Recently the methods and practices developed have been employed in diverse areas of geosciences, with Carrassi et al. (2018); Vetra-Carvalho et al. (2018) providing recent overviews. Further DA has seen a huge expansion into other scientific disciplines with applications in, for example, robotics (Berquin and Zell, 2022), economic modelling (Nadler et al., 2019) and plasma physics (Sanpei et al., 2021). In the era of digital twinning, which involves combining high-fidelity representations of reality with the optimal use of observations, real-time data has become vital and DA frameworks have naturally been incorporated.

The area of data learning has also emerged where DA approaches are integrated with machine learning techniques (Buizza et al., 2022).

There are various popular DA techniques, with variational methods (3DVar and 4DVar) (Thépart et al., 1993) and *ensemble Kalman filter*s (EnKFs) (Evensen, 1994; Burgers et al., 1998) being extensively used in operational and research settings. Bannister (2017) provides a good overview of operational methods. However, these methods have difficulties with handling non-linear problems and with representing uncertainties accurately (Lei et al., 2010; Bocquet et al., 2010). For instance Miyoshi (2005); Kondo and Miyoshi (2019) updated an ensemble of $10\,240$ particles using the EnKF to demonstrate the bimodality of some distributions due to inherent nonlinearities. Furthermore, the continuing growth in compute hardware performance has allowed running increasingly complex and high resolution models which are able to resolve non-linear processes happening at a fine spatial scale (Vetra-Carvalho et al., 2018), creating an increasing demand for DA methods which are able to accurately quantify uncertainty in such settings. *Particle filter*s (PFs) (Gordon et al., 1993) are an alternative approach which offer the promise of consistent DA for problems with non-linear dynamics and non-Gaussian noise distributions. Traditionally the main difficulty with particle filtering techniques has been the 'curse of dimensionality' (Bengtsson et al., 2008; Bickel et al., 2008; Snyder, 2011), where in high dimensional settings filtering leads to degeneracy of the importance weights associated with each particle and loss of diversity within an ensemble unless the ensemble size scales exponentially with the observation dimension. To improve the applicability of PFs there have been many recent developments involving: localization techniques (e.g., the reviews in Farchi and Bocquet (2018); Graham and Thiery (2019)), incorporation of tempering/mutation steps (e.g., Cotter et al. (2020); Ruzayqat et al. (2022)), hybrid approaches, improved computational implementations and combination of the above with improved proposal distributions. The above ongoing efforts have extended the applicability of PF methods within geoscientific domains. Leeuwen et al. (2019) provides an overview on the integration of particle filters in high-dimensional geoscience applications.

The DA paradigm of optimally combining observations and model has a wide range of applications. However, an existing hurdle impeding the incorporation of data into pre-existing dynamical models is the lack of readily available software packages capable of bridging the two sources of information: model and observations. This is the motivation behind ParticleDA.jl, to provide a generic and user-friendly framework to enable the incorporation of particle filtering techniques with pre-existing numerical models. ParticleDA.jl is an open-source package in Julia which provides efficient implementations of several particle filter algorithms, and also importantly offers an extensible framework to allow the simple addition of new filter implementations. It has been developed to be agnostic to the forward model to ensure applicability in a wide range of settings and emphasis has been placed on computational efficiency and scalability on high-performance computing systems. Initial efforts have been focused on the integration with spatially dependent numerical models, however the implementation is applicable to a much more general class of state space models (see Sect. 2) allowing incorporation in a broad range of applications.

For a specific class of state space models with additive Gaussian state and observation noise, and linear observation operators, ParticleDA.jl allows particle filtering with the so-called 'locally optimal' proposal distribution. Though the latter is not amongst the latest contributions in the PF literature (e.g., Doucet et al. (2000)), it has not been extensively explored in the high-dimensional *partial differential equation* (PDE) driven systems which we use as case studies in this work. As noted in

| Package name | Algorithms | Parallelism |
|---|---|---|
| DataAssim.jl (Barth et al., 2016) | EnKF, 4DVar | |
| EnKF.jl (Le Provost, 2016) | EnKF | |
| Kalman.jl (Schauer et al., 2018) | KF | |
| KalmanFilters.jl (Schoenbrod, 2018) | KF | |
| LowLevelParticleFilters.jl (Carlson et al., 2018) | PF, KF | Shared memory |
| ParticleFilters.jl (Sunberg et al., 2017) | PF | |
| SequentialMonteCarlo.jl (Lee and Piibeleht, 2017) | PF (SMC) | Shared memory |
| **ParticleDA.jl** | PF, KF | Shared and distributed memory |

**Table 1.** Summary of algorithms implemented and parallelism support in existing Julia data assimilation packages.

Snyder (2011), improved proposals used within PFs can in practice significantly improve the performance of the algorithm, but by themselves do not overcome the curse of dimensionality. Our numerical experiments illustrate that ParticleDA.jl can already be usefully applied in practice to models with moderately high dimensions, however, an important line of future work will be
extending the framework with additional filter implementation incorporating approaches such as localization and tempering to allow scaling to very high-dimensional settings.

Within the Julia ecosystem, there are several existing packages which implement data assimilation algorithms. DataAssim.jl (Barth et al., 2016) provides implementations of a range of EnKF and extended *Kalman filter* (KF) methods and an incremental variant of 4DVar. EnKF.jl (Le Provost, 2016) implements stochastic and deterministic (square-root) variants of the EnKF which
can be combined with various approaches (e.g. covariance inflation) to avoid ensemble collapse in models with deterministic dynamics. Kalman.jl (Schauer et al., 2018) and KalmanFilters.jl (Schoenbrod, 2018) both provide implementations of the exact KF algorithm for linear Gaussian models, with KalmanFilters.jl additionally implementing unscented variants of the KF for use in models with non-linear dynamics or observation operators. ParticleFilters.jl (Sunberg et al., 2017) and LowLevelParticleFilters.jl (Carlson et al., 2018) both provide PF implementations, with LowLevelParticleFilters.jl additionally providing
KF implementations. SequentialMonteCarlo.jl (Lee and Piibeleht, 2017) provides an interface for implementing (and example implementations of) the wider class of *sequential Monte Carlo* (SMC) methods, of which PFs can be considered a special case, with the ability to run particle ensembles in parallel on multiple threads. EnsembleKalmanProcesses.jl (Dunbar et al., 2022) implements several derivative-free optimization algorithms based on the EnKF mainly targeted at Bayesian inverse problem settings. Another package to note in the Julia ecosystem is DataAssimilationBenchmarks.jl (Grudzien and Bocquet, 2022;
Grudzien et al., 2022) which offers a framework to empirically validate and develop novel DA techniques.

Table 1 summarizes the algorithm and parallelism support of existing Julia data assimilation packages along with our package ParticleDA.jl. The existing packages, LowLevelParticleFilters.jl and SequentialMonteCarlo.jl, which support parallelization of operations across ensemble members both use shared-memory parallelism, with tasks run simultaenously across multiple threads on the same device. In contrast, as described in Sect. 3.3, our package ParticleDA.jl supports both shared and distributed
memory parallelism which enables efficient deployment on *high performance computing* (HPC) systems.

We implemented ParticleDA.jl in the Julia programming language (Bezanson et al., 2017) because of its combination of performance and productivity, which enables rapid prototyping and development of high-performance numerical applications (Churavy et al., 2022; Giordano et al., 2022), with the possibility of using both shared and distributed memory parallelism strategies. In particular, Julia makes use of the multiple-dispatch programming paradigm, which is particularly well-suited for designing a program which combines different models with different filtering algorithms, keeping the two concerns separated. This allows domain experts and software engineers to collaborate on the code using the same high-level language.

The rest of this manuscript is organized as follows. The mathematical set-up of the particle filtering algorithm is defined in Sect. 2 along with the various filtering proposal distributions implemented. Section 3 outlines the code structure and parallelization schemes. Sections 4 and 5 illustrate applications of the framework to simple low-dimensional state space models, namely a stochastically driven damped simple harmonic oscillator model and a stochastic variant of the Lorenz '63 chaotic attractor model (Lorenz, 1963). Section 6 introduces an application to a spatially extended state space model, specifically a tsunami modelling test case formulated as a linear Gaussian state space model, with validation of the filtering approaches and parallel performance scaling results. In Sect. 7 the incorporation with a more complex non-linear atmospheric dynamical model is investigated along with some results. Finally in Sect. 8 concluding remarks and future work are outlined.

## 2  Particle filtering

Let $\boldsymbol{x}_t \in \mathbb{R}^{d_x}$ represent the state of the model at an integer time index $t$ and $\boldsymbol{y}_t \in \mathbb{R}^{d_y}$ the vector of observations of the system at this time index. We assume a state space model formulation, with the states following a Markov process and the observations depending only on the state at the corresponding time index, that is

$$\boldsymbol{x}_0 \sim p_0(\cdot); \quad \boldsymbol{x}_t \sim p_t(\cdot \mid \boldsymbol{x}_{t-1}), \quad \boldsymbol{y}_t \sim g_t(\cdot \mid \boldsymbol{x}_t), \quad t \geq 1; \tag{1}$$

where $p_0 : \mathbb{R}^{d_x} \to \mathbb{R}_{\geq 0}$ is the density of the initial state distribution, $p_t : \mathbb{R}^{d_x} \times \mathbb{R}^{d_x} \to \mathbb{R}_{\geq 0}$, $t \geq 1$ are the densities of the state transition distributions and $g_t : \mathbb{R}^{d_y} \times \mathbb{R}^{d_x} \to \mathbb{R}_{\geq 0}$, $t \geq 1$ are the densities of the conditional distribution of the observations given the current states.

A key assumption of the state space model formulation is that the state of the system evolves stochastically in time. In geophysical applications, commonly the models of interest are specified as the solution to time-dependent PDEs or systems of *ordinary differential equation*s (ODEs), for which the state dynamics are inherently deterministic. Without a stochastic element to the dynamics the evolution of the state over time is entirely determined by the initial state. To perform particle filtering in such models we must therefore augment the deterministic dynamics of the model with stochastic updates. These stochastic updates can be considered as random forcings of the model representing physical processes not modelled in the deterministic model as well as the discretization errors introduced when simulating ODE and PDE models (Leeuwen et al., 2019). Importantly the stochastic updates should maintain any constraints or relationships between the state variables in the underlying physical phenomena being modelled – for example for state vectors corresponding to spatial discretizations of a continuous field, the stochastic updates should maintain any assumed smoothness properties of the field.

For the most part, we will concentrate on a specialization of this general state space model class, whereby the state and observations are both subject to additive Gaussian noise and the observations depend linearly on the state, which covers a wide range of modelling scenarios in practice. Concretely we consider a state update of the form

$$\boldsymbol{x}_t = F_t(\boldsymbol{x}_{t-1}) + \boldsymbol{u}_t, \quad \boldsymbol{u}_t \sim \mathcal{N}(0, Q), \quad t \geq 1, \tag{2}$$

where $F_t : \mathbb{R}^{d_x} \to \mathbb{R}^{d_x}$ is the forward operator at time index $t$, representing the deterministic dynamics of the system and $\boldsymbol{u}_t \in \mathbb{R}^{d_x}$ is the additive Gaussian state noise at time index $t$, representing stochastic aspects of the system dynamics. The observations are modelled as being generated according to

$$\boldsymbol{y}_t = H(\boldsymbol{x}_t) + \boldsymbol{v}_t, \quad \boldsymbol{v}_t \sim \mathcal{N}(0, R), \quad t \geq 1, \tag{3}$$

where $H \in R^{d_y \times d_x}$ is a linear observation operator and $\boldsymbol{v}_t \in \mathbb{R}^{d_y}$ is the additive Gaussian observation noise. The distributions of the state and observation noise are parameterized by positive-definite covariance matrices $Q \in \mathbb{R}^{d_x \times d_x}$ and $R \in \mathbb{R}^{d_y \times d_y}$ respectively.

The objective of the particle filter is to estimate the filtering distribution for each time index $t$ which is the conditional probability distribution of the state $\boldsymbol{x}_t$ given observations $\boldsymbol{y}_1, ..., \boldsymbol{y}_t$ up to time index $t$, with the density of the filtering distribution at time index $t$ denoted $\pi_t(\boldsymbol{x}_t | \boldsymbol{y}_{1:t})$.

The particle filtering algorithm builds on sequential importance sampling by introducing additional resampling steps. See Doucet et al. (2000) for an in-depth introduction but the key features are introduced in Algorithm 1. An ensemble of *particles* $\{\boldsymbol{x}_t^{(i)}\}_{i=1}^N$ represents an approximation to the filtering distribution at each time index $t \geq 1$ as $\pi_t(\mathrm{d}\boldsymbol{x}_t | \boldsymbol{y}_{1:t}) \approx \frac{1}{N} \sum_{i=1}^N \delta_{\boldsymbol{x}_t^{(i)}}(\mathrm{d}\boldsymbol{x}_t)$. In each filtering step, new values for the particles are sampled from a proposal distribution (more details about the proposals implemented in ParticleDA.jl are given in Sect. 2.1) and importance weights computed for each proposed particle value. At the end of the filtering step, the weighted proposed particle ensemble is resampled to produce a new uniformly weighted ensemble to use as the input to the next filtering step.

## 2.1 Proposal distributions

Two forms of proposal distributions are implemented in ParticleDA.jl: the 'naive' bootstrap proposal, applicable to general state space models described by Eq. (1), and the 'locally optimal' proposal, which can be tractably computed only for a restricted class of state space models, including importantly those described by Eqs. (2) and (3).

The bootstrap proposal ignores the observations with the particle proposals sampled from the state transition distributions,

$$q_t(\boldsymbol{x}_t | \boldsymbol{x}_{t-1}, \boldsymbol{y}_t) = p_t(\boldsymbol{x}_t | \boldsymbol{x}_{t-1}), \tag{4}$$

with the unnormalized importance weights at time index $t \geq 1$ then simplifying to

$$W_t(\boldsymbol{x}_t, \boldsymbol{x}_{t-1}, \boldsymbol{y}_t) = g_t(\boldsymbol{y}_t | \boldsymbol{x}_t) \ (= W_t(\boldsymbol{x}_t, \boldsymbol{y}_t)). \tag{5}$$

While appealingly simple and applicable to a wide class of models, the bootstrap particle filter performs poorly when observations are informative about the state due to the observations being ignored in the proposal. In such cases the proposed particles

**Algorithm 1** Particle filter

---

1: Initialize particles $\{\boldsymbol{x}_0^{(i)}\}_{i=1}^N$, with $\boldsymbol{x}_0^{(i)} \sim p_0(\cdot)$, for $1 \le i \le N$.

2: **for** time index $t = 1$ **to** $T$ **do**

3:     **for** particle index $i = 1$ **to** $N$ **do**

4:         Sample proposed particle $\tilde{\boldsymbol{x}}_t^{(i)} \sim q_t(\cdot \,|\, \boldsymbol{x}_{t-1}^{(i)}, \boldsymbol{y}_t)$.

5:         Compute (unnormalized) importance weight $w_t^{(i)} = W_t(\tilde{\boldsymbol{x}}_t^{(i)}, \boldsymbol{x}_{t-1}^{(i)}, \boldsymbol{y}_t) = \frac{p_t(\tilde{\boldsymbol{x}}_t^{(i)} \,|\, \boldsymbol{x}_{t-1}^{(i)}) g_t(\boldsymbol{y}_t \,|\, \tilde{\boldsymbol{x}}_t^{(i)})}{q_t(\tilde{\boldsymbol{x}}_t^{(i)} \,|\, \boldsymbol{x}_{t-1}^{(i)}, \boldsymbol{y}_t)}$.

6:     **end for**

7:     Generate new (equally weighted) particles $\{\boldsymbol{x}_t^{(i)}\}_{i=1}^N$ by resampling from the weighted empirical distribution

$$\boldsymbol{x}_t^{(i)} \sim \frac{\sum_{i=1}^N w_t^{(i)} \delta_{\tilde{\boldsymbol{x}}_t^{(i)}}(\cdot)}{\sum_{i=1}^N w_t^{(i)}}$$

8: **end for**

---

will typically be far away from the mass of the true filtering distribution, with the importance weights in this setting tending to have high variance leading to *weight degeneracy* whereby all the normalized importance weights but one are close to zero.

To alleviate the tendency to weight degeneracy we can use alternative proposal distributions which decrease the variance of the importance weights. For proposals distributions $q_t(\boldsymbol{x}_t \,|\, \boldsymbol{x}_{t-1}, \boldsymbol{y}_t)$ which condition only on the previous state $\boldsymbol{x}_{t-1}$ and current observation $\boldsymbol{y}_t$, the optimal proposal, in the sense of minimising the variance of the importance weights, can be shown (Doucet et al., 2000) to be

$$q_t(\boldsymbol{x}_t \,|\, \boldsymbol{x}_{t-1}, \boldsymbol{y}_t) = \frac{p_t(\boldsymbol{x}_t | \boldsymbol{x}_{t-1}) g_t(\boldsymbol{y}_t \,|\, \boldsymbol{x}_t)}{\int p_t(\tilde{\boldsymbol{x}}_t | \boldsymbol{x}_{t-1}) g_t(\boldsymbol{y}_t \,|\, \tilde{\boldsymbol{x}}_t) \mathrm{d}\tilde{\boldsymbol{x}}_t}, \tag{6}$$

with corresponding unnormalized importance weights

$$W_t(\boldsymbol{x}_t, \boldsymbol{x}_{t-1}, \boldsymbol{y}_t) = \int p_t(\tilde{\boldsymbol{x}}_t | \boldsymbol{x}_{t-1}) g_t(\boldsymbol{y}_t \,|\, \tilde{\boldsymbol{x}}_t) \mathrm{d}\tilde{\boldsymbol{x}}_t \ (= W_t(\boldsymbol{x}_{t-1}, \boldsymbol{y}_t)). \tag{7}$$

Note that in this case the importance weights are independent of the sampled values of the particle proposals.

For general state space models, sampling from this *locally optimal proposal* and computing the importance weights can be infeasible due to the integral in Eq. (6) and Eq. (7) not having a closed form solution. However, for the specific case of a state space model of the form described by Eqs. (2) and (3), the proposal distribution has the tractable form

$$q_t(\boldsymbol{x}_t \,|\, \boldsymbol{x}_{t-1}, \boldsymbol{y}_t) = \mathcal{N}\left(\boldsymbol{x}_t \,|\, F_t(\boldsymbol{x}_{t-1}) + QH^\mathsf{T}(HQH^\mathsf{T} + R)^{-1}(\boldsymbol{y}_t - HF_t(\boldsymbol{x}_{t-1})), Q - QH^\mathsf{T}(HQH^\mathsf{T} + R)^{-1}HQ\right), \tag{8}$$

with corresponding importance weights

$$W_t(\boldsymbol{x}_t, \boldsymbol{x}_{t-1}, \boldsymbol{y}_t) = \mathcal{N}\left(\boldsymbol{y}_t \,|\, HF_t(\boldsymbol{x}_{t-1}), HQH^\mathsf{T} + R\right). \tag{9}$$

To generate samples from the locally optimal proposal distribution, we exploit that for $\tilde{\boldsymbol{x}}_t \sim \mathcal{N}(F_t(\boldsymbol{x}_{t-1}), Q)$ and $\tilde{\boldsymbol{y}}_t \sim \mathcal{N}(H\tilde{\boldsymbol{x}}_t, R)$ – that is $(\tilde{\boldsymbol{x}}_t, \tilde{\boldsymbol{y}}_t)$ sampled from the joint distribution on the state and observation given the previous state $\boldsymbol{x}_{t-1}$

under the state space model – then

$$\boldsymbol{x}_t = \tilde{\boldsymbol{x}}_t + QH^{\mathsf{T}}(HQH^{\mathsf{T}} + R)^{-1}(\boldsymbol{y}_t - \tilde{\boldsymbol{y}}_t), \tag{10}$$

is distributed according to the locally optimal proposal distribution in Eq. (8). Importantly this means that to use the locally optimal proposal distribution when filtering we only need to implement functions for sampling from the state transition and observation models, and functions for evaluating the matrix terms in Eq. (10) – that is $QH^{\mathsf{T}}$ and $HQH^{\mathsf{T}} + R$. Note that unlike a direct implementation of sampling from Eq. (8) by performing a Cholesky factorization of the proposal covariance matrix, we do *not* need to explicitly evaluate or store a $d_x \times d_x$ covariance matrix, and only need to perform a $\mathcal{O}(d_y^3)$ linear solver and $\mathcal{O}(d_x d_y)$ matrix-vector multiplication rather than a $\mathcal{O}(d_x^3)$ Cholesky decomposition. As well as reducing the time and memory complexity of the linear algebra operations, this approach reduces the implementation burden on a user wishing to apply the locally optimal proposal, by reusing functions required for simulating the forward model, and ensures any algorithmic efficiencies used in the implementation of simulating the forward model are also leveraged in sampling from the locally optimal proposal distribution.

In both the bootstrap and locally optimal proposal, the stochastic nature of the state transitions are essential to maintaining diversity in to the ensemble, ensuring any particles duplicated in the previous resampling step give rise to distinct proposals. For state space models with state update and observation model described by Eqs. (2) and (3) specifically, we can see that the bootstrap and locally optimal proposals converge to the same degenerate distribution $\delta_{F_t(\boldsymbol{x}_{t-1})}$ as the state noise vanishes, that is $Q \to 0$. This emphasises the importance of using stochastic state dynamics for the PF algorithms used here to remain valid.

## 2.2 Resampling

A vital aspect of all PF algorithms is the resampling step shown in Algorithm 1. Resampling multiplies particles found at good positions in space that agree with observations and removes unwanted particles, concentrating computational effort on the more plausible ensemble members. It is key for establishing analytical results showing that Monte-Carlo errors in estimates of expectations under the filtering distribution are controlled *uniformly* in time, see, e.g., the standard reference Del Moral (2004). Such a result provides a critical justification for the powerful performance of PF-based algorithms in many applications. ParticleDA.jl implements a systematic resampling scheme (Douc and Cappé, 2005), which uses a single uniform random variate to resample all the particle indices.

A useful metric for capturing the variability of the weights before resampling, and indicating whether weight degeneracy has occurred, is the estimated *effective sample size* (ESS), which is defined as

$$\mathrm{ESS}_t = \frac{\left\{\sum_{i=1}^{N} w_t^{(i)}\right\}^2}{\sum_{i=1}^{N}\{w_t^{(i)}\}^2}. \tag{11}$$

where $\{w_t^{(i)}\}_{i=1}^{N}$ are the *unnormalized* particle importance weights. The estimated ESS approximates the number of independent samples that would produce estimates of similar variance as the ones obtained by the available (correlated) particles.

**Algorithm 2** Structure of ParticleDA.jl `run_particle_filter` function

---

1: Initialize model
2: Initialize states ($\parallel$)
3: Initialize filter
4: Initialize summary statistics ($\parallel$)
5: **for** time index $t = 1$ **to** $T$ **do**
6:     Sample proposal and compute (unnormalized) particle weights for current observation $\boldsymbol{y}_t$ ($\parallel$)
7:     Gather particle weights ($\leftrightarrow$)
8:     Normalize particle weights ($\circ$)
9:     Resample particle indices according to weights ($\circ$)
10:    Broadcast new particle indices ($\leftrightarrow$)
11:    Copy particle states according to new indices ($\leftrightarrow$)
12:    Update summary statistics ($\leftrightarrow, \parallel$)
13:    Write outputs ($\circ$)
14: **end for**

---

## 3   Code structure

As previously stated, ParticleDA.jl is designed to be forward model agnostic (*i.e.* capable of running with arbitrary state space models). To enable this the model and filter portions of ParticleDA.jl are carefully delineated. The high-level structure of the main `run_particle_filter` function used to perform filtering with a state space model given a sequence of observations $\boldsymbol{y}_1, \ldots, \boldsymbol{y}_T$ is summarized in Algorithm 2, where $T$ is the total number of observation times. Operations which use thread-based parallelism are labelled with ($\parallel$). When run across multiple processes, operations involving communication across ranks are labelled with ($\leftrightarrow$) and those which only run on the coordinating rank are labelled with ($\circ$). Further details on the filter and model interface and parallelization implementation are given in the following sections.

### 3.1   Filter interface

Implementing a filtering algorithm in ParticleDA.jl requires providing implementations of two functions, `init_filter` and `sample_proposal_and_compute_log_weights!`, with the corresponding methods dispatched on a filter type argument which is a concrete subtype of the `ParticleFilter` abstract type. Implementations are currently provided for particle filters with bootstrap proposals (with corresponding type `BootstrapFilter`) and locally optimal proposals (with corresponding type `OptimalFilter`).

The `init_filter` method deals with initializing any filter specific data structures, including allocating arrays to hold the particle weights, resampling indices and ensemble summary statistics. A set of shared filter parameters are passed to the `init_filter` method as an instance of a dedicated `FilterParameters` type, with functionality provided for reading these parameters from a YAML file. Key parameters include the number of particles in the ensemble, number of tasks used

| Function name | Description |
|---|---|
| `get_state_dimension` | Get value of $d_x$ |
| `get_observation_dimension` | Get value of $d_y$ |
| `sample_initial_state!` | Sample $\boldsymbol{x}_0 \sim p_0(\cdot)$ |
| `sample_observation_given_state!` | Sample $\boldsymbol{y}_t \sim g_t(\cdot \mid \boldsymbol{x}_t)$ |
| `get_log_density_observation_given_state` | Get value of $\log g_t(\boldsymbol{y}_t \mid \boldsymbol{x}_t)$ |
| `update_state_deterministic!` `update_state_stochastic!` | Sample $\boldsymbol{x}_t \sim p_t(\cdot \mid \boldsymbol{x}_{t-1})$ |
| `get_observation_mean_given_state!` | Get value of $H\boldsymbol{x}_t$ ⋆ |
| `get_covariance_state_noise` | Get value of $Q_{i,j}$ ⋆ |
| `get_covariance_observation_noise` | Get value of $R_{i,j}$ ⋆ |
| `get_covariance_state_observation_given_previous_state` | Get value of $(HQ)_{i,j}$ ⋆ |
| `get_covariance_observation_observation_given_previous_state` | Get value of $(HQH^{\mathsf{T}} + R)_{i,j}$ ⋆ |

**Table 2.** Summary of main functions defining model interface. Functions in rows marked ⋆ only required when using locally optimal proposal.

when scheduling parallelizable operations in multi-threaded code segments (see Sect. 3.3), the seed for the pseudo random number generator used to generate random variates during filtering and the file path to write filtering outputs to, as well as options for controlling the verbosity of the filter output.

The `sample_proposal_and_compute_log_weights!` method provides an implementation of generating new values for an ensemble of particles from the proposal distribution associated with the filter type and computing the corresponding unnormalized particle weights (in particular their logarithms to maintain numerical stability), corresponding to respectively lines 4 and 5 in Algorithm 1. As hinted by the presence of an `!` suffix in the function name, a convention in Julia for indicating functions which mutates one or more of its arguments, the `sample_proposal_and_compute_log_weights!` function computes the updates to the arrays representing the particle states and weights in place. The proposal generation and weight computation stages are combined into a single rather than two separate functions, to allow the filter implementation to avoid redundant computations of quantities required in both sampling the proposals and computing the weights. For example, both the locally optimal proposal distribution in Eq. (8) and corresponding weights in Eq. (9) require the values of the particles after the deterministic update $F_t$ but before addition of the state noise.

### 3.2 Model interface

To support filtering in general state space models while still allowing filters to exploit additional structure in the model when present, we define an extensible model interface, with a core set of functions requiring implementation for all state space models, with model classes with additional structure able to extend this core interface. In particular we exploit this approach for conditionally Gaussian state space models having a state update and observation models of the form described by Eq. (2) and Eq. (3) respectively, to allow filtering using the locally optimal proposal distribution in Eq. (8). Table 2 summarizes the key

functions requiring implementation both within the core interface for general state space models, and for the restricted class of models for which the locally optimal proposal can be applied, along with a brief description of what operations they perform in the notation of Sect. 2.

A key pair of functions in Table 2 are `update_state_deterministic!` and `update_state_stochastic!`, which for general state space models when applied in sequence correspond to sampling from the state transition distribution $x_t \sim p_t(\cdot \,|\, x_{t-1})$, while for models with state updates of the specific form in Eq. (2), correspond respectively to updating the state by applying the deterministic forward operator $F_t$ and incrementing by a state noise vector $u_t \sim \mathcal{N}(0, Q)$. While the state transitions for general state space models may not factor into a composition of deterministic and stochastic updates, full generality is still maintained as `update_state_deterministic!` can leave the state vector unchanged (corresponding to an identity operation) with `update_state_stochastic!` then solely responsible for sampling from the state transition distribution.

The functions `get_covariance_state_observation_given_previous_state` and `get_covariance_observation_observation_given_previous_state` are required for computing the locally optimal proposal update in Eq. (10). In both cases the models' implementations of these functions are required to evaluate a scalar covariance value for a pair of integer state or observation indices in $1{:}d_x$ and $1{:}d_y$ respectively. Providing the state noise covariance $Q_{ij}$ can be evaluated in $\mathcal{O}(1)$ (with respect to $d_x$ and $d_y$) operations for pairs of state indices $i, j \in 1{:}d_x$, which will typically be the case if the state noise corresponds to the discretisation of a Gaussian random field with an explicit covariance function, and the observation operator $H$ is sparse (for example each observation depends on the state at only one or a few indices), then both functions can be evaluated in $\mathcal{O}(1)$ cost. The $d_y \times d_x$ and $d_y \times d_y$ matrices $HQ$ and $HQH^{\mathsf{T}} + R$ are then evaluated by calling the functions for grids of state and observation indices, with $\mathcal{O}(d_y d_x)$ and $\mathcal{O}(d_y^2)$ costs respectively, without requiring explicit construction of the $d_x \times d_x$ matrix $Q$, which is particularly important when $d_x$ is large.

As a further optimization, models can implement a `get_state_indices_correlated_to_observations` function which returns a subset of the state indices $1{:}d_x$ which excludes indices for which the corresponding state variable is uncorrelated to all observation variables (that is $i \in 1{:}d_x$ such that $(HQ)_{ij} = 0$ for all $j \in 1{:}d_y$). For example, if the state corresponds to the discretisation of a set of spatial fields with the observations corresponding to only one of these fields, then only the state indices corresponding to the observed field would need to be returned. This is then used to avoid needing to compute the known zero covariance terms.

The functions `get_covariance_observation_noise` and `get_covariance_state_noise` are not required directly for the locally optimal proposal update. However, typically they will be used in the definitions of other model methods and models are required to provide implementations to allow testing the model methods for internal consistency, and to support a Kalman filter implementation for linear-Gaussian models.

As well as providing implementation of the function in Table 2, models are required to implement an initialization function which is passed to the top-level `run_particle_filter` function and used to initialize an instance of the model data structure type the model interface functions are dispatched on. This initialization function is passed a dictionary of model specific parameter values read from a YAML file, and the number of tasks that may be simultaneously scheduled when running

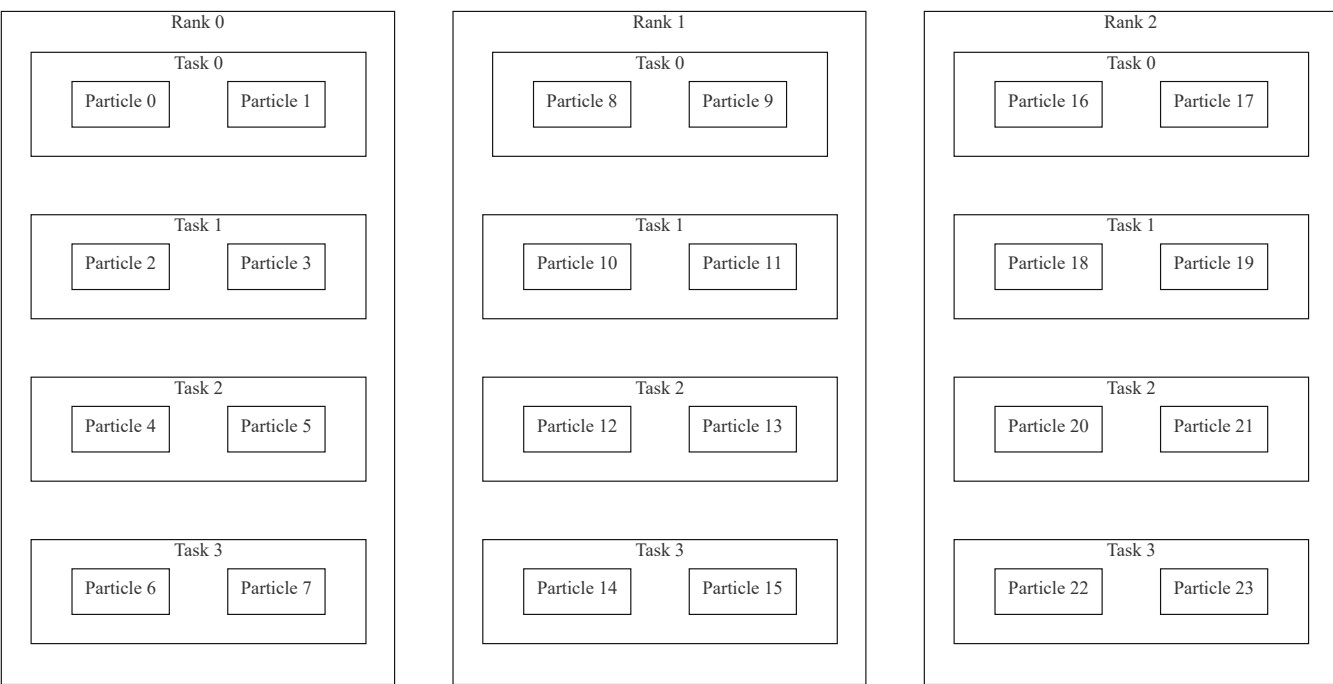

**Figure 1.** Visualization of the hierarchical parallelization model in ParticleDA.jl for an example case where updates to 24 particles are distributed across 3 MPI ranks. Each rank splits the particles across 4 tasks, with these tasks scheduled to run in parallel across 2 or more threads on each rank.

model functions in parallel (see Sect. 3.3), allowing the assignment of per-task buffers for use in computing intermediate results
265   while remaining thread-safe.

### 3.3   Parallelization scheme

As stated both shared and distributed memory parallelization approaches can be leveraged within ParticleDA.jl, to exploit both multiple processing elements sharing memory on a single node (for example *central processing unit* (CPU) cores) and multiple nodes (potentially each with multiple processing elements) in a cluster. Particle and weight updates are parallelized across
270   multiple threads on shared memory systems using the native task-based multi-threading support in Julia. In distributed memory environments ParticleDA.jl allows parallelizing across processes (ranks) with communication between processes performed using the Julia package MPI.jl  (Byrne et al., 2021), which acts as a wrapper around a *message passing interface* (MPI) implementation installed on the system. MPI.jl has been found to be able to achieve little to no overhead in applications with thousands of MPI ranks  (Giordano et al., 2022). ParticleDA.jl uses HDF5 files for file based input (of the observed data used
275   for filtering) and output (of statistics computed during filtering), using the HDF5.jl Julia package; when running in distributed setting file input-output is performed only on a single coordinating rank.

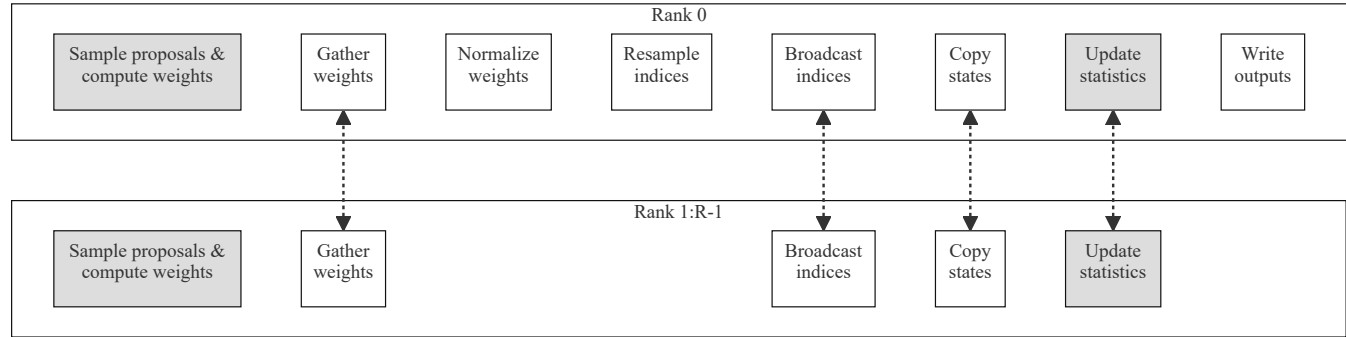

**Figure 2.** An overview of how the key stages in the filtering loop are distributed across $R$ ranks in ParticleDA.jl. Rank 0 is the coordinating rank which mediates communication across ranks and performs file input and output. Shaded nodes indicate stages which are run in parallel across multiple threads on each rank. Edges between nodes indicate stages which involve communication across ranks.

An illustration of how per-particle operations are distributed in the two-level parallelization scheme is illustrated in Fig. 1. Each MPI rank is assigned an equal proportion of the total number of particles in the ensemble. Within each MPI rank, operations which can be parallelized across particles are scheduled across multiple *tasks* each associated with a subset of the particles assigned to the rank. The tasks are run simultaneously across multiple threads, with the flexibility in number of tasks per rank allowing a trade off between improved load balancing across processing elements on a rank by having multiple tasks scheduled per parallel thread, and the increased overhead involved in scheduling more tasks.

A sketch of the key operations in the main filtering loop and how they are distributed across multiple ranks is shown in Fig. 2. A key principle is to reduce as much as possible the requirement to communicate the full particle state vectors between ranks. Particles remain local to specific ranks for all operations other than when copying states as part of the resampling step, with this step potentially requiring particles with large weight which are duplicated after resampling to be copied point-to-point to other ranks. Communication between ranks is also required when gathering the unnormalized particle weights to the coordinating rank to allow normalization and when broadcasting the resampled particle indices from the coordinating rank to other ranks, however these operations only require communicating a single scalar per particle.

Communication between ranks is also required when computing any summary statistics of the estimated filtering distributions at each time index. ParticleDA.jl currently supports estimating the mean and, optionally, the variance of the filtering distributions for each state dimension, with a summary statistic type argument to the top-level `run_particle_filter` function allowing specification of which summary statistics to compute. Sufficient statistics of the local particles for the relevant summary statistics are computed on each rank, before these local sufficient statistics are accumulated on the coordinating rank using an MPI reduce operation and used to compute the statistics of interest. For CPU architectures for which MPI.jl supports using custom reductions[1], a more numerically stable 'pooled' algorithm (Chan et al., 1982) is used for computing the mean and variance (adapting the example code given in Byrne et al. (2021)); implementations of the less numerically stable

---

[1]https://github.com/JuliaParallel/MPI.jl/issues/404.

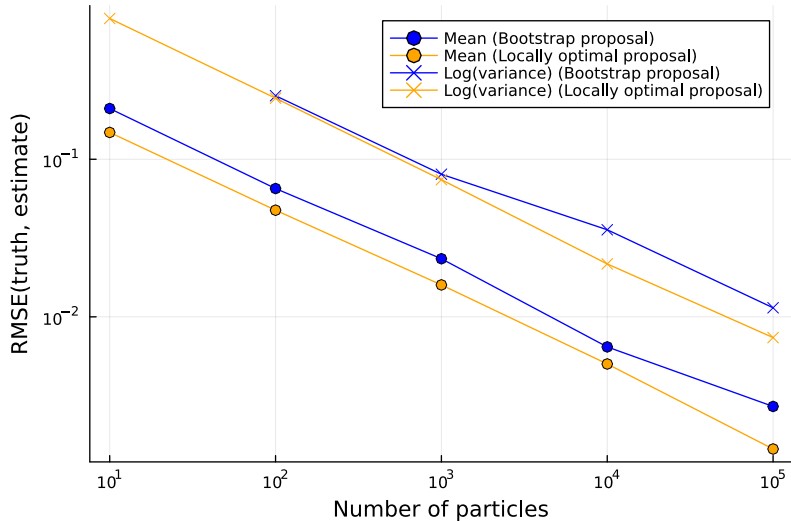

**Figure 3.** RMSE in particle filter estimates of filtering distribution means and (log) variances against number of particles for the damped simple harmonic oscillator model. The RMSE values are calculated against ground truth values computed using a Kalman filter and are computed for the mean of the squared errors across all state components and time steps.

'naive' algorithms which directly accumulates the sum and sum of squares, which can be performed using standard MPI sum reductions, are also provided as a fallback for running on other CPU architectures.

## 4 Stochastically driven damped simple harmonic oscillator

As a tractable first test case we consider a two-dimensional state space model corresponding to the time discretization of a stochastic differential equation

$$\mathrm{d}\boldsymbol{x}(\tau) = \begin{pmatrix} 0 & 1 \\ -\omega_0^2 & -\omega_0/Q \end{pmatrix} \boldsymbol{x}(\tau)\mathrm{d}\tau + \begin{pmatrix} 0 \\ 1 \end{pmatrix} \mathrm{d}W(\tau),$$

representing a damped simple harmonic oscillator driven by a Wiener noise process $W(\tau)$, with $\tau$ the (continuous) time coordinate, $\omega_0$ the frequency of the undamped oscillator and $Q$ a quality factor for the oscillator. This process has been proposed as a model for astronomical time series data (Foreman-Mackey et al., 2017), with details of its formulation as a state space model given in Jordán et al. (2021). Importantly, the state space model is linear-Gaussian so we can use a Kalman filter to exactly compute the true Gaussian filtering distributions.

We use an instance of the model with parameters $\omega_0 = 1$ and $Q = 2$. We assume an observation model $\boldsymbol{y}_t \sim \mathcal{N}(\boldsymbol{x}_t, 0.5^2 I)$, with $\boldsymbol{x}_t = \boldsymbol{x}(0.2t)$ (that is a fixed time step 0.2 between observation times), and simulate observations from the model for $T = 200$ time steps with initial state distribution $\boldsymbol{x}_0 \sim \mathcal{N}(\boldsymbol{0}, I)$. Fig. 3 shows the *root mean squared error*s (RMSEs) in particle filter estimates of the means and log-variances of the Gaussian filtering distributions (compared to ground truth values computed

using a Kalman filter), as a function of the number of particles used in the ensemble, for filters using both the bootstrap and locally optimal proposal. We see that the locally optimal proposal gives a small but consistent improvement in RMSE for a given ensemble size, reflecting the lower variance in the empirical estimates to the filtering distributions. As expected the errors in the filter estimates appear to be asymptotically tending to zero at a polynomial rate in the ensemble size, providing some assurance of the correctness of the ParticleDA.jl filter implementations.

## 5 Lorenz system


The Lorenz 63 system was introduced by Lorenz (1963) and is a non-linear dynamical model capturing simplified representation of thermal convection. The model is defined by the ODE system

$$\frac{\mathrm{d}x_1(\tau)}{\mathrm{d}\tau} = \sigma(x_2(\tau) - x_1(\tau)), \qquad \frac{\mathrm{d}x_2(\tau)}{\mathrm{d}\tau} = \rho x_1(\tau) - x_2(\tau) - x_1(\tau)x_3(\tau), \qquad \frac{\mathrm{d}x_3(\tau)}{\mathrm{d}\tau} = x_1(\tau)x_2(\tau) - \beta x_3(\tau), \qquad (12)$$

where $\tau$ is the time coordinate, $x_1(\tau), x_2(\tau)$ and $x_3(\tau)$ are the prognostic variables of the model and $\sigma$, $\rho$ and $\beta$ are free
parameters. As outlined by Lorenz (1963) we have set the free parameters to $\sigma = 10$, $\rho = 28$ and $\beta = \frac{8}{3}$ as this set up will lead to chaotic behaviour. To formulate as a state space model with state transitions of the form described by Eq. (2), we set $F_t$ to the *flow map* corresponding to numerically solving the initial value problem for the ODE system in Eq. (12) over a fixed inter-observation time interval of 0.1 time units such that $\boldsymbol{x}_t = (x_1(0.1t), x_2(0.1t), x_3(0.1t))$ and use additive isotropic state noise with covariance $Q = 0.5^2 I$. The Tsit5 solver with adaptive time-stepping from the Julia package DifferentialEquations.jl
(Rackauckas and Nie, 2017) is used to solve the ODE system. The initial state distribution is taken to be $\boldsymbol{x}_0 \sim \mathcal{N}(\boldsymbol{0}, 0.5^2 I)$. We assume an observation model $\boldsymbol{y}_t \sim \mathcal{N}(\boldsymbol{x}_t, 1.0^2 I)$ and simulate observations for $T = 500$ times from the model to use for filtering.

Fig. 4 illustrates the performance of a filtering run with $N = 100$ particles on the simulated observations using the locally optimal proposal. The left subplot shows the (noisy) observations and estimated mean of the filtering distributions at each of the
observation times, note the appearance of the Lorenz attractor. The right subplot shows the evolution of the RMSE calculated for the estimated mean against the observations for different number of particles ($N$) with the locally optimal proposal.

### 5.1 Nonlinear observation operator

The effect of a nonlinear observation operator on the performance of the bootstrap filter proposal is explored, note that the locally optimal proposal can not be readily applied in this set up and therefore only the bootstrap filter is used. Two observation
operators are introduced: the linear $H(\boldsymbol{x}_t) = \boldsymbol{x}_t$ and the nonlinear $H(\boldsymbol{x}_t) = \log |\boldsymbol{x}_t|$ case.

Simulations of the Lorenz system (Eq. 12) are carried out with a similar set up as in section 5, an initial state distribution taken to be $\boldsymbol{x}_0 \sim \mathcal{N}(\boldsymbol{0}, 0.5^2 I)$, an observation model $\boldsymbol{y}_t \sim \mathcal{N}(H(\boldsymbol{x}_t), 1.0^2 I)$ and additive isotropic state noise with covariance $Q = 0.5^2 I$. The bootstrap filter is used and the observations are assimilated every 0.1 time units. For the non-linear case the observation operator acts upon the observation values which are then perturbed by draws from the independent observation
error.

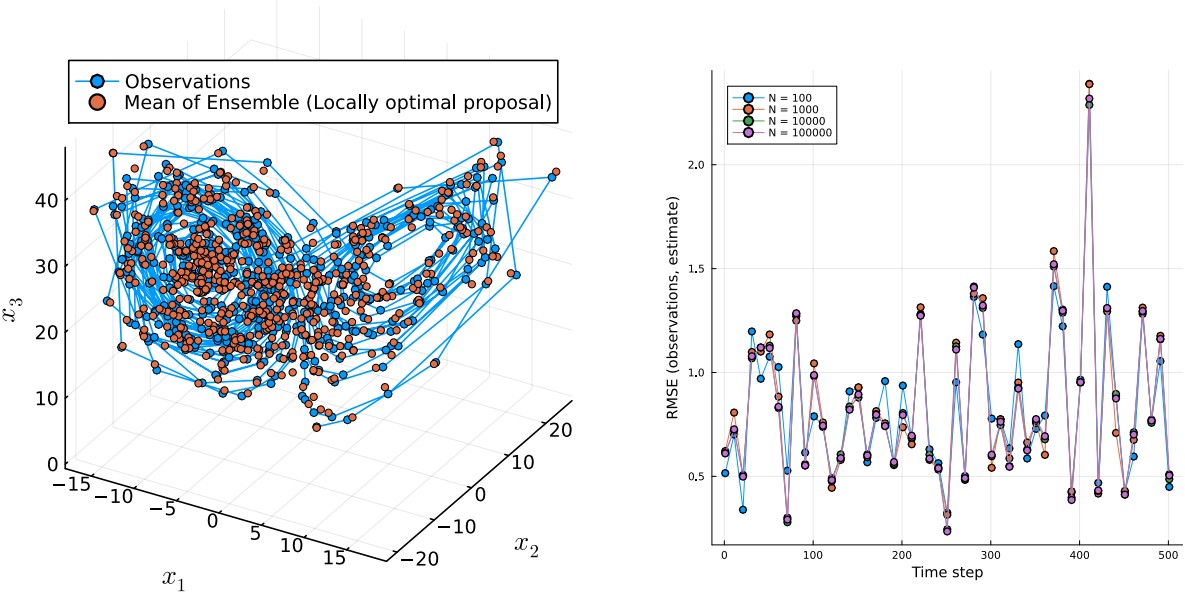

**Figure 4.** Left: Mean of the particles and the observations at each time step in state space for $N = 100$. Right: The RMSE calculated through time for different number of particles ($N$).

The system is simulated for $T = 5000$ time steps and a time averaged RMSE is calculated for the last $4500$ steps for the linear observation, the non-linear observation and a no assimilation case. The time averaged RMSE results are plotted in 5. The effect of the non-linear observation operator can be clearly seen, where the time averaged RMSE for the linear case is lower in all set-ups. However, the non-linear observation operator leads to a lower error when compared to the no assimilation case.

## 6 Tsunami model

As a more complex test case, we now consider a tsunami modelling example. Tsunamis are rare events which have the capacity of causing severe loss of life and damages. At present, tsunami warning centres rely on crude decision matrices, pre-computed databases of high resolution simulations or 'on-the-fly' real-time simulations to rapidly deduce the hazard associated with an event (Gailler et al., 2013). These existing approaches have been developed with seismically generated tsunamis in mind and the alternative tsunamigenic sources (landslide and volcanic eruptions) are less well constrained. The ongoing efforts of incorporating data assimilation techniques within tsunami modelling could augment warning centres capability in this regard (Maeda et al., 2015; Gusman et al., 2016). It should be noted that the tsunami model built into ParticleDA.jl is a drastic simplification to tsunami models used in operational practice but provides a useful test case for users and showcases the potential of particle filters within tsunami modelling efforts. Further, it allows for direct validation with a Kalman filter as the resulting state space model is linear-Gaussian.

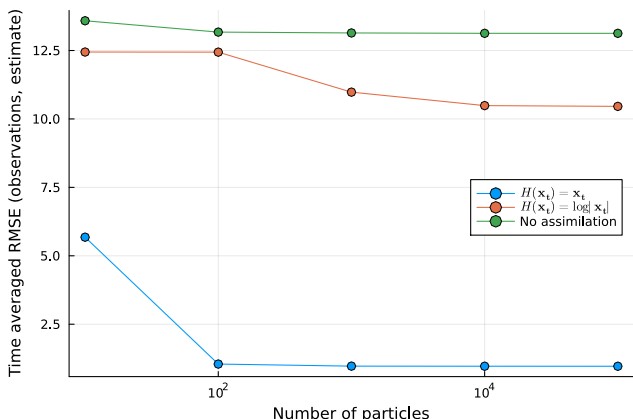

**Figure 5.** Time averaged RMSE for three different runs with various number of particles: a naive ensemble (no assimilation), the linear observation case $H(\boldsymbol{x}_t) = \boldsymbol{x}_t$ and the non-linear observation case $H(\boldsymbol{x}_t) = \log|\boldsymbol{x}_t|$.

As a first order approximation, two-dimensional linear long-wave equations (Goto, 1984), corresponding to a linearization of the shallow water equations, are used to capture the tsunami dynamics. Assuming a state space model with state transitions of the form described in Eq. (2), the (linear) deterministic forward operator $F_t(\boldsymbol{x}_t)$ for the test case is defined by numerically solving the PDEs

$$\frac{\partial \eta(\tau, s_1, s_2)}{\partial \tau} = -\frac{\partial u(\tau, s_1, s_2)}{\partial s_1} - \frac{\partial v(\tau, s_1, s_2)}{\partial s_2},$$

$$\frac{\partial u(\tau, s_1, s_2)}{\partial \tau} = -gh(s_1, s_2)\frac{\partial \eta(\tau, s_1, s_2)}{\partial s_1},$$

$$\frac{\partial v(\tau, s_1, s_2)}{\partial \tau} = -gh(s_1, s_2)\frac{\partial \eta(\tau, s_1, s_2)}{\partial s_2}, \tag{13}$$

where $\tau$ is the time coordinate, $(s_1, s_2)$ are the spatial coordinates, $\eta(\tau, s_1, s_2)$ is the free surface elevation (wave height), $h(s_1, s_2)$ is the (static) water depth, $g$ is the acceleration due to gravity and $u(\tau, s_1, s_2)$ and $v(\tau, s_1, s_2)$ are the components of the depth averaged horizontal velocities. The system of linear PDEs in Eq. (13) is solved using a first order finite difference scheme with absorbing boundary conditions, using a Julia reimplementation of the *tsunami data assimilation code* (TDAC)

accompanying Gusman et al. (2016).

The state vector $\boldsymbol{x}_t$ is defined as the concatenation of the flattened vectors formed by the spatial discretizations of the fields $\eta$, $u$ and $v$ on a $51 \times 51$ uniform grid over a square spatial domain $[0, 2 \times 10^5] \times [0, 2 \times 10^5]$ (resulting in an overall state dimension $d_x = 3 \times 51^2 = 7803$), with uniform interval of 2 time units between observation times, that is

$$\boldsymbol{x}_t = \big(\eta(2t, 0, 0), \eta(2t, 0, 4 \times 10^3), \ldots, \eta(2t, 0, 2 \times 10^5), \eta(2t, 4 \times 10^3, 0), \ldots, \eta(2t, 2 \times 10^5, 2 \times 10^5),$$
$$u(2t, 0, 0), u(2t, 0, 4 \times 10^3), \ldots, u(2t, 0, 2 \times 10^5), u(2t, 4 \times 10^3, 0), \ldots, u(2t, 2 \times 10^5, 2 \times 10^5),$$
$$v(2t, 0, 0), v(2t, 0, 4 \times 10^3), \ldots, v(2t, 0, 2 \times 10^5), v(2t, 4 \times 10^3, 0), \ldots, v(2t, 2 \times 10^5, 2 \times 10^5)\big).$$

The additive state noise is chosen as the spatial discretizations of independent Gaussian random fields for each of the variables $\eta$, $u$ and $v$, with a Matérn covariance kernel with length scale parameter $\lambda = 500$, smoothness parameters $\mu = 2.5$ and marginal standard deviation parameter $\sigma = 0.01$ used for all three fields. The spatially correlated nature of the state noise distribution ensures the perturbed spatial fields remain smooth. A circulant embedding method (Dietrich and Newsam, 1997) implemented in the Julia package GaussianRandomFields.jl (Robbe, 2017) is used to efficiently simulate Gaussian random fields on a

uniform grid using fast Fourier transforms, resulting in a $\mathcal{O}(d_x \log d_x)$ operation cost complexity for each realisation. For filtering the initial state distribution is also chosen to correspond to a zero-mean Gaussian distribution corresponding the spatial discretizations of independent Gaussian random fields for each of the variables $\eta$, $u$ and $v$, with a Matérn covariance kernel with the same parameters $(\lambda, \mu, \sigma)$ as above.

We assumed noisy point-wise observations of the free surface elevation field $\eta$ at 15 'station' locations $\{s_1^{(m)}, s_2^{(m)}\}_{m=1}^{15}$,

chosen as grid points randomly sampled from a uniform distribution over the spatial grid for simplicity, with independent observation noise with standard deviation 0.01, that is $\boldsymbol{y}_t \sim \mathcal{N}\left((\eta(2t, s_1^{(m)}, s_2^{(m)}))_{m=1}^{15}, 0.01^2 I\right)$. For the simulation of the observations, to produce an initial wave producing perturbation, the mean of the initial state distribution for the free surface elevation components is altered to correspond to the function

$$\bar{\eta}_0(s_1, s_2) = \begin{cases} \left((1 + \cos(\pi(s_1 - a)/c))(1 + \cos(\pi(s_2 - a)/c))\right) d/4 & (s_1 - a)^2 + (s_2 - a)^2 \leq c^2, \\ 0 & \text{otherwise,} \end{cases} \tag{14}$$

evaluated at the grid points with $a = 10^4$, $b = 10^4$, $c = 3 \times 10^4$, $d = 30$, with the mean of the velocity components left as zero. As the initial state distribution assumed when filtering differs we therefore have a small degree of model mismatch. The observations are simulated for $T = 640$ times, with the PDE system numerically integrated in time for 4 time steps of 0.5 time units between each pair of observation times.

## 6.1    Validation

We performed an initial filtering run on the simulated observations using an ensemble of $N = 50$ particles using the locally optimal proposals. Snapshots of the simulated free surface elevation field used to generate the observations and corresponding particle estimate of the mean of the filtering distribution on the free surface elevation field are shown in Fig. 6.

As the tsunami state space model implemented here is linear-Gaussian a Kalman filter was used to compute ground truth values for the means (and covariances) of the filtering distributions, and these were then compared to the filtering estimates for

various ensemble sizes $N$ and proposal distributions. Fig. 7 (left) shows the RMSE in the estimate of the filtering distribution mean for each observation time time for PF using both the locally optimal and bootstrap proposals with the same ensemble size ($N = 50$). The filter using the locally optimal proposal can be observed to give a consistent improvement in the accuracy of the filtering distribution estimates across time. Fig. 7 (right) instead shows the RMSE in the estimate of the filtering distribution mean at a single observation time $\tau = 200$, for filtering runs with varying ensemble sizes $N$ for both bootstrap and locally

optimal proposals, the results indicate a consistent gain in accuracy of the filtering estimates when using the locally optimal compared to bootstrap proposal, across a range of different ensemble sizes $N$.

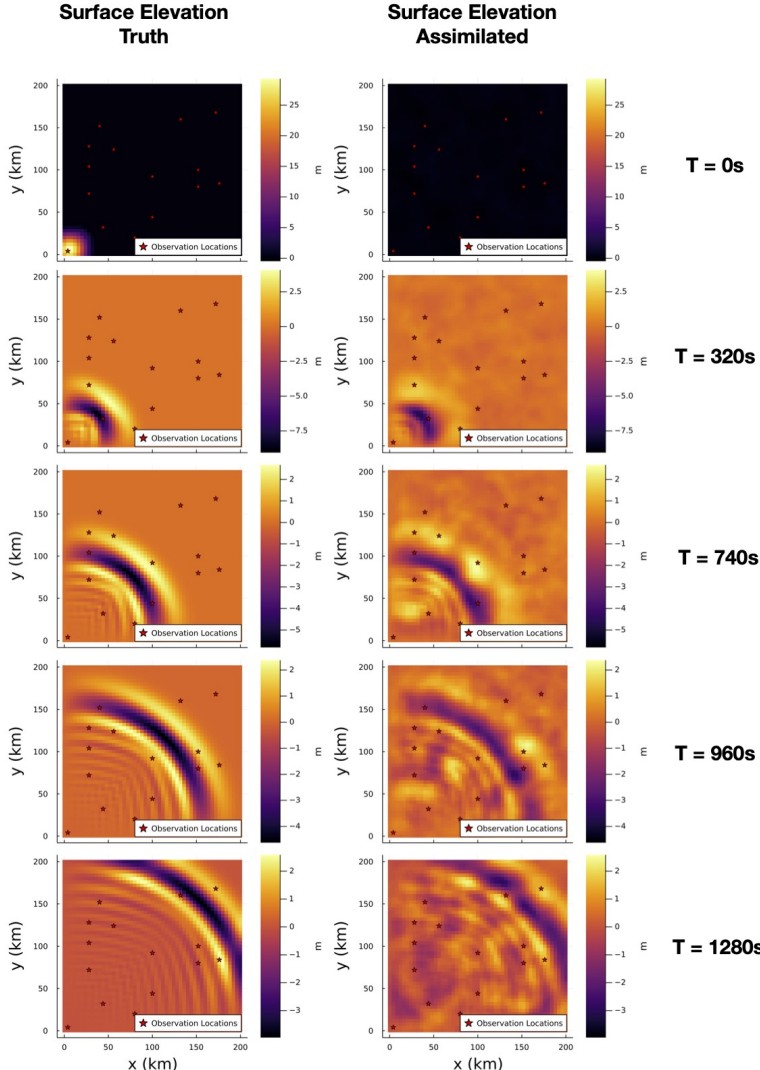

**Figure 6.** Snapshots of the surface elevation used to generate the simulated observations (left) and the corresponding estimated filtering distribution means using $N = 50$ particles (right).

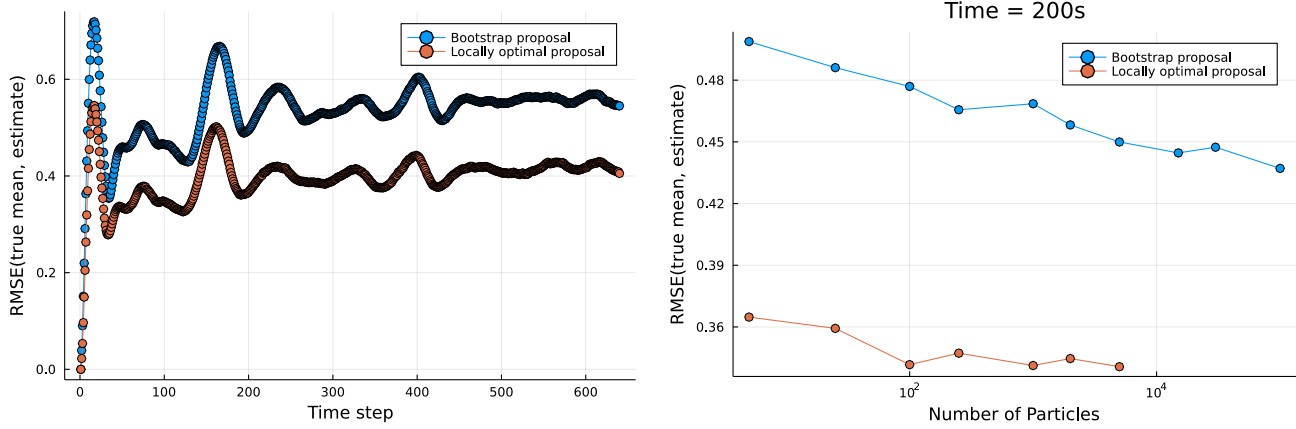

**Figure 7.** Left: RMSE in particle filter estimates of filtering distribution across observation times for the tsunami models with a fixed ensemble size of $N = 50$ for filters using both bootstrap and locally optimal proposals. Right: RMSEs in particle filter estimates of the filtering distribution mean for both the bootstrap and locally optimal proposal at $\tau = 200$ for varying ensemble size $N$. Note: The RMSE values are calculated against the true mean of the filtering distributions coming from a Kalman filter run.

## 6.2 Parallelization performance

As discussed in Sect. 3.3 ParticleDA.jl is capable of leveraging both shared and distributed parallelism. Scaling runs on ARCHER2, which is the UK's Tier-1 supercomputer, have been carried out to highlight the performance in practice. A weak
scaling study, using the same experimental set up as described in section 6.1 is run with the bootstrap proposal keeping the number of particles per core constant while increasing the number of nodes. The compute nodes on ARCHER2 consist of $2 \times$ AMD EPYC 7742, 2.25 GHz, 64-core, with 8 *non uniform memory access* (NUMA) regions per node (16 cores per NUMA region, 8 cores per *core complex die* (CCD) and 4 cores per *core complex* (CCX) (shared L3 cache)). The weak scaling runs try to optimize for this hardware architecture with various runs targeting a MPI rank per NUMA / CCD / CCX region and an
appropriate number of threads per MPI rank (Fig. 8). The weak scaling efficiency is defined as $E(N) = \frac{T(2)}{T(N)}$, where $T(N)$ is the wall time for running on $N$ MPI ranks. There are 2 particles per core so at the maximum number of cores (2048) and ranks (128) tested here there are 4096 particles.

As stated in Sect. 3.3 the main performance bottleneck are the communication steps: the copying of states and the gathering of particle weights which require point-to-point communications and a global communication step respectively. Another com-
ponent which contributes to poor scaling for large node counts is updating the summary statistics, which requires a reduction of the mean and optionally the variance for each state dimension over all MPI ranks. For the results presented here we have mostly remedied this loss of performance by collecting statistics at the final filtering iteration only. However, it should be noted that for cases which need frequent outputted statistics this will contribute to a degradation of the parallel performance.

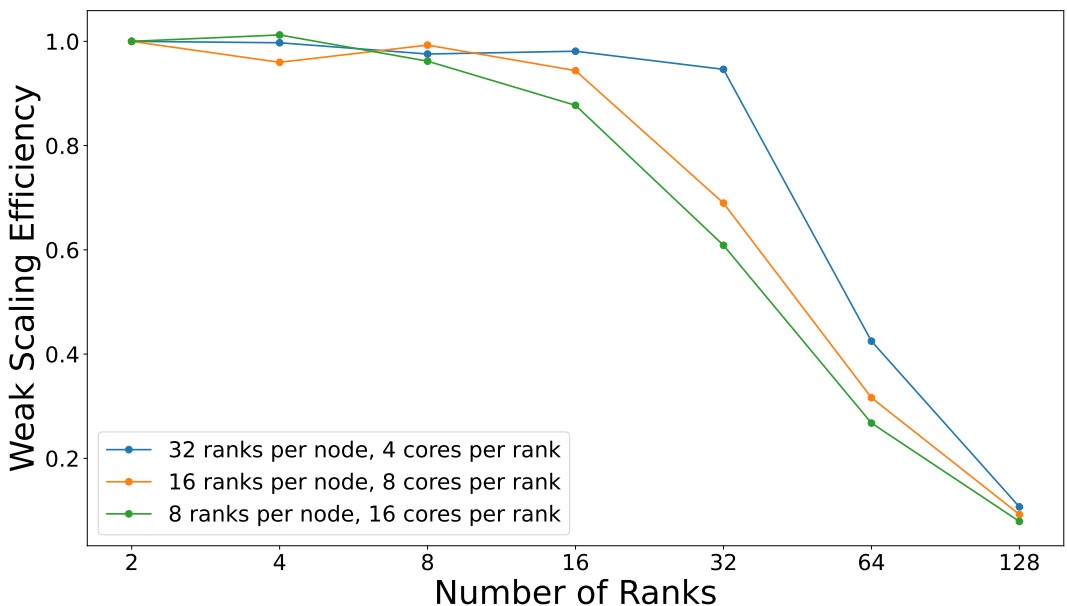

**Figure 8.** Weak scaling parallel efficiency for the tsunami model test case with different set ups of ranks per node and cores per rank on ARCHER2. A drop off in performance can be seen when moving from single to multi-node runs.

## 7 Atmospheric general circulation model (AGCM)

An integration of ParticleDA.jl with an atmospheric dynamical model, *simplified parameterizations primitive equation dynamics* (SPEEDY), showcases the efforts involved in coupling the software with pre-existing model implementations. SPEEDY is an AGCM which was developed by Molteni (2003) and consists of a spectral primitive-equation dynamic core along with a set of simplified physical parameterization schemes. The SPEEDY model retains the core characteristics of the current state-of-the-art AGCMs but requires drastically less (orders of magnitude) computational resources (Molteni, 2003). This computational efficiency allows one to utilize the model to carry out large ensemble and/or data assimilation experiments. According to Molteni (2003) the SPEEDY model accurately simulates the general structure of global atmospheric circulation and exhibits similar systematic errors to the state-of-the-art AGCM, albeit with larger error amplitudes. The model implementation used here (Hatfield, 2018) is written in Fortran and provides an interesting example of the integration steps required to interface with ParticleDA.jl. The coupling with ParticleDA.jl relies on the SPEEDY implementation being set up to output its data fields at set intervals.

As stated SPEEDY is a simplified AGCM model. The prognostic variables consist of the zonal and meridional wind velocity components $(u, v)$, temperature $(T)$, specific humidity $(q)$ and surface pressure $(p_s)$. A T30 resolution of the model is used here which corresponds to a horizontal grid size of $96 \times 48$ with 8 vertical layers. The vertical layers are defined by sigma levels, where the pressure is normalized by the surface pressure $(p/p_s)$.

We extend the deterministic SPEEDY model to a state space model setting, by using a state transition update of the form described by Eq. (2), with numerical simulation of the SPEEDY model forward in time by 6 simulated hours corresponding to the deterministic forward operator $F_t$. The state vector $\boldsymbol{x}_t$ is defined as the concatenation of the flattened vectors corresponding to the spatial discretizations of the prognostic variable fields $u$, $v$, $T$, $q$ and $p_s$, with each of the first four variables being defined in three dimensions across a $96 \times 48 \times 8$ spatial grid, while the final surface pressure variable $p_s$ is defined in two dimensions on a $96 \times 48$ spatial grid, resulting in an overall state dimension of $d_x = 4 \times 96 \times 48 \times 8 + 96 \times 48 = 152\,064$.

The additive Gaussian state noise is assumed to correspond to spatial discretizations of independent two-dimensional Gaussian random fields for the surface pressure $p_s$ and for each vertical level for the prognostic variables $u$, $v$, $T$ and $q$. To reflect the underlying spherical geometry over which the spatial grid is defined, a non-stationary covariance function using a Matérn kernel on the geodesic (great-circle) distance between the points on the sphere the grid points correspond to is used, with the Matérn kernels using common values of $\lambda = 1$ and $\mu = 2.5$ for the length scale and smoothness parameters respectively, while the marginal standard deviation parameter $\sigma$ is set separately for each prognostic variable, with $\sigma = 1$ for $u$, $v$ and $T$, $\sigma = 0.001$ for $q$ and $\sigma = 100$ for $p_s$. The GaussianRandomFields.jl package is again used to generate realizations of the (spatially discretized) random fields, with the use of a non-stationary covariance function in this case necessitating an approach which uses an eigendecomposition of the full $4608 \times 4608$ covariance matrix for each discretized two-dimensional field to generate the samples. The geodesic distance based covariance function used is not guaranteed to be positive definite which is heuristically dealt with by setting all negative eigenvalues to zero. We recognize that this approach of introducing state noise into the dynamics has some limitations but for the purposes here is sufficient.

The data assimilation experiments carried out here followed a similar set-up to that used in Miyoshi (2005). A linear-Gaussian observation model of the form described by Eq. (3) is used, with observations assumed to be available only for the surface pressure field (in this regard differing from the set-up used by Miyoshi (2005)) at 50 spatial locations corresponding to randomly sampled grid points, with additive independent observation noise with standard deviation 10hPa. The initial state used to generate the simulated observations is generated by performing a one simulated year 'spin-up' of the deterministic SPEEDY model from a resting atmosphere ($u = v = 0$) initial condition, with simulated observations then generated for 250 observation times at 6 hourly intervals using the state space model. Initial state values for a filtering run using $N = 256$ particles and locally optimal proposals were generated by performing a long-term (10 simulated years) run of the deterministic SPEEDY model, with the state selected randomly from the simulated times in the final month of simulation and state noise of the same distribution used in state transitions added.

## 7.1 Results

In Fig. 9 snapshots of the true surface pressure (top left) and the ensemble estimate of the mean surface pressure (top right) after 250 assimilation cycles are shown. Minimal differences can be observed. The sub-plots in the bottom row showcase the time averaged $L_2$ error for ensemble mean estimates with and without assimilation. The $L_2$ error is calculated against the true surface pressure fields used to simulate the observations at each grid cell over the 250 assimilation cycles. The time

averaged errors are dominated by mid-latitude patterns but the ensemble run without assimilation exhibits larger errors. This error comparison validates that the assimilation is giving improved estimates of the state of the system.

As stated in the introduction, one of the key benefits of particle filters is to provide the promise of non-linear and non-Gaussian DA. To highlight this sample distributions of the surface pressure at various observation locations at different time points are shown in Fig. 10. The distributions across the $N = 256$ particles exhibit heavy tails towards the true surface pressure at the given locations. It should be noted that similar non-Gaussian distributions were showcased by (Miyoshi et al., 2014; Kondo and Miyoshi, 2019) in a near identical experimental set-up but with an ensemble Kalman filter. However, a key differ-

ence to be highlighted here is the relative size of the ensembles used, 256 particles here versus a 10,240 ensemble size used by (Miyoshi et al., 2014) to generate the non-Gaussian distributions.

## 8    Conclusions

We have developed a flexible Julia package, ParticleDA.jl, for performing particle-filter based data assimilation, with the potential of offering improved filtering accuracy when working with models exhibiting non-Gaussianity in the filtering distributions.

The use of a high-level language Julia, both simplifies the process for users wanting to apply the package to their own models and for developers wishing to extend the package with new filter implementations, while still maintaining similar computational efficiency to lower level compiled languages like Fortran and C++.

    Particular attention has been paid to ensuring ParticleDA.jl is suitable for performing filtering on HPC systems, with a versatile two-level model used to support both shared and distributed memory parallelism. This is important in allowing efficient

exploitation of the typically complex hierarchies of processing elements used in modern HPC systems (see for example the description of the hardware architecture of ARCHER2 in Section 6.2), both when running large ensembles of models where each particle can be simulated on a single processing element, but also for the perhaps more practically relevant setting of running smaller ensembles of more complex models which require multiple processing elements to simulate a single particle.

    ParticleDA.jl currently provides implementations of particle filters using bootstrap and locally optimal proposals, with the

former applicable to general state space models and the latter to a more restricted subset of state space models with Gaussian state transition distributions and linear-Gaussian observation models. As illustrated in our numerical experiments, particle filters using the locally optimal proposal distributions can offer significantly improvements in the accuracy of filtering estimates for a given ensemble size where applicable. However, as noted in the introduction, particle filters using the locally optimal proposal distribution are known to still suffer from a 'curse of dimensionality' requiring the ensemble size to scale exponentially

with the system dimension to avoid weight degeneracy (Snyder, 2011). An important future extension to ParticleDA.jl will therefore be in providing implementations of filtering algorithms exploiting approaches such as spatial localization (Farchi and Bocquet, 2018) to allow scaling to very high dimensional geophysical applications. Implementations of filters exploiting spatial localization will be necessarily applicable to a restricted subclass of spatially extended state space models; similar to the approach used for implementing the locally optimal proposal filter, the extensible nature of the model interface in ParticleDA.jl

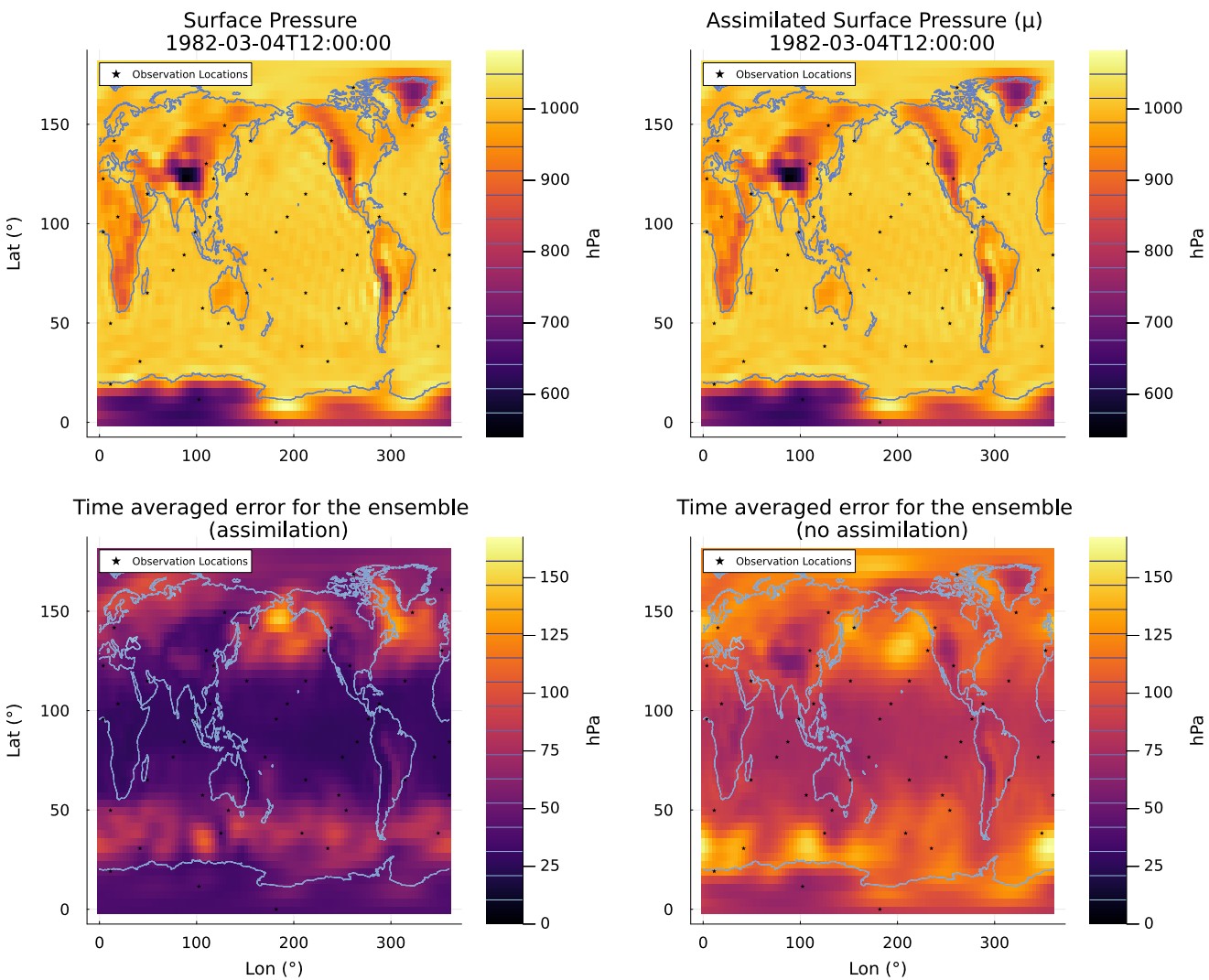

**Figure 9.** Top Row: Snapshot of the true surface pressure (left) and mean assimilated surface pressure (right) ($N = 256$) after 250 assimilation cycles (12:00:00 04/03/1982 UTC). Bottom Row: Time averaged $L_2$ error for the mean of the assimilation run (left) and for the mean of an ensemble run without assimilation (right). The 50 observation locations are highlighted by the black stars.

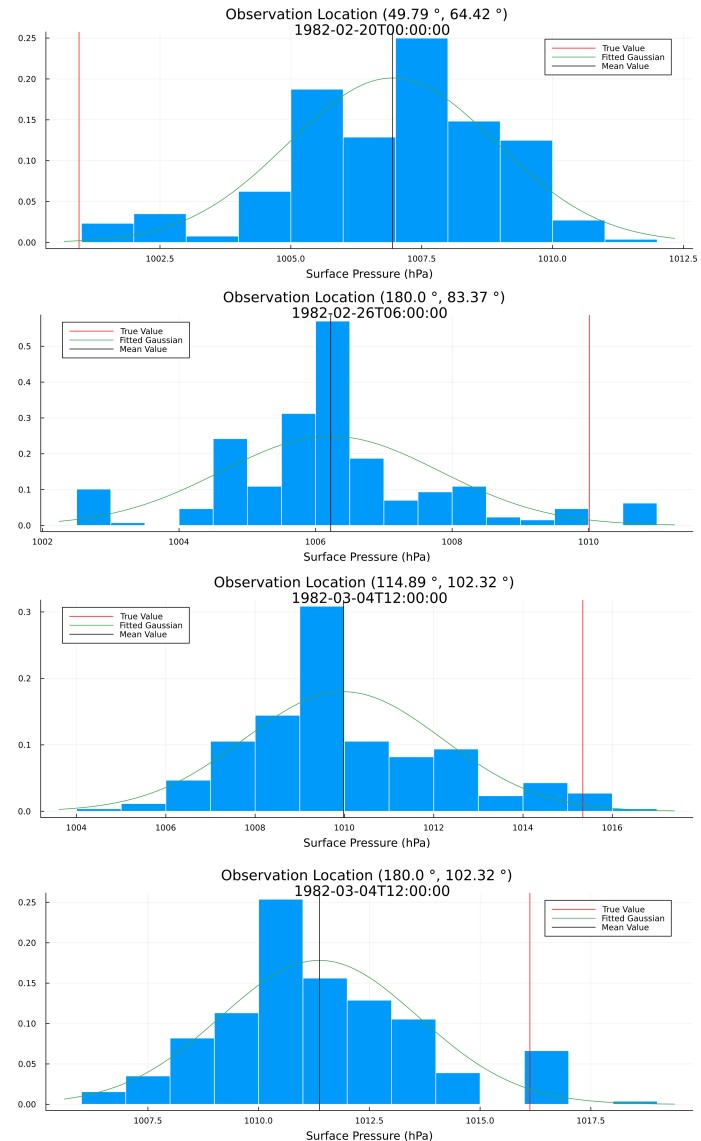

**Figure 10.** Normalized histograms corresponding to the estimates of the marginal filtering distributions of the surface pressure at various observation locations and at different points in time from a filtering run with an ensemble of $N = 256$ particles. The true surface pressures are highlighted by the vertical red line and the mean surface pressure of the particles are highlighted by the black vertical line. The green line represents a fitted Gaussian distribution.

should allow model agnostic localized filter implementations to be added by simply defining additional functions required to be implemented by the model interface.

Another key element that a user should be aware of is the generation of state noise and the role that it plays (Evensen et al., 2022). For some geophysical applications this can be a non-trivial task as the definition of the state noise should respect the smoothness of the state variable and any underlying physical constraints. For example, particular efforts have been made in the AGCM case (section 7) to capture the underlying spherical geometry by implementing a non-stationary covariance function using a Matérn kernel based on the geodesic distance between points on the sphere. This approach of introducing state noise into the dynamics offers an improvement to a stationary covariance function but still has limitations and does not guarantee that physical constraints are conserved. This example highlights the important role of the state noise and potential users should be aware of the efforts needed to accurately capture this in their systems of interest.

Overall, the aim of our platform is to enable easily accessible and accurate, fast DA for a wide range of users. We hope that various scientific communities will adopt ParticleDA.jl, possibly leading to fast step-changes in some geoscientific investigations and beyond.

*Code availability.* The code is freely available at https://github.com/Team-RADDISH/ParticleDA.jl

*Author contributions.* DG lead the computations and applications. TK, MMG and MG created the Julia platform, its I/O and computational acceleration. AB led the design of the locally optimal proposal implementation. SG directed the overall research.

*Competing interests.* There are no competing interests at present

*Acknowledgements.* The RADDISH (Real-time Advanced Data assimilation for Digital Simulation of numerical twins on HPC) project supported this research. RADDISH was part of The Tools, Practice and Systems programme of the AI for Science and Government (ASG), UKRI's Strategic Priorities Fund awarded to the Alan Turing Institute, UK (EP/T001569/1). We also acknowledge funding for this research from UKAEA (T/AW085/21) for the project Advanced Quantification of Uncertainties In Fusion modelling at the Exascale with model order Reduction (AQUIFER).

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
