# Peer review of "ParticleDA.jl v.1.0: A distributed particle filtering data assimilation package"

_Geoscientific Model Development, 2023_

## Referee Comment (RC1)

**Review of gmd-2023-38: *ParticleDA.jl v.1.0: A real-time data assimilation software platform**

By Daniel Giles, Matthew M. Graham, Mosè Giordano, Tuomas Koskela, Alexandros Beskos, and Serge Guillas

13 Apr 2023

Verdict: **Major Revision**

In this paper the authors introduce a new data assimilation software framework, ParticleDA.jl. This uses the particle filter algorithm, which is among the class of ensemble data assimilation algorithms and in principle has the best general performance in nonlinear settings where Gaussianity cannot be assumed. They have developed this software framework in Julia in a modular way which makes it easy to integrate it to arbitrary nonlinear dynamical systems, and they demonstrate its applicability to a series of models of increasing complexity. The authors have also focused somewhat on the high-performance capability of their package, and demonstrate weak scaling performance up to 16 nodes, though I would consider this just a start and tests with much higher node counts are required. The paper was easy to read and provides sufficient detail to reproduce the experiments. I would recommend acceptance after the below major comment is addressed, which pertains to further verification that their assimilation methodology is working correctly for the highest complexity model considered, SPEEDY. I am looking forward to reading their response.

**Major Comments**

- Line 312: My only major comment was triggered by this statement: "It can be seen that the areas of greatest percentage error coincide with areas that lack observation stations." I want to challenge this statement as I don't see such a strong coincidence. Instead the error patterns seem to be dominated by patterns of midlatitude weather systems, which is what you would expect to see if you compared two snapshots of surface pressure in the same model run at different times. In fact, I don't see sufficient evidence that the assimilation is actually working. It probably is, but it's hard to tell unless you compare also with a run without any assimilation. A number of further plots would help to make this section more complete:
    1. A snapshot of surface pressure for the truth run some time into the run (after the assimilation error has asymptoted).
    2. A corresponding snapshot of the surface pressure averaged across the particle ensemble (or a randomly chosen particle, as you prefer — they should be similar if the ensemble has mostly converged on the truth). This should be close to plot 1. if assimilation is working.
    3. A corresponding snapshot of the surface pressure for a run without any assimilation (just a single run) to demonstrate how much divergence would be expected.

4. To complete the 4-plot square, you could also show an error plot. I would suggest to produce a plot like Fig. 7 right, but averaged over time. This should filter out the midlatitude variability and actually show an error pattern correlated with the observation locations.

Note that surface pressure is an odd variable to plot due to the dominance of mountainous areas. You might prefer to use 850 hPa temperature or something else with fewer regional nonuniformities. The above is just a guide and I leave it to the authors to demonstrate that their assimilation methodology is working correctly for the SPEEDY model.

**Minor Comments**

- Line 65: typo — "targeted".
- Line 169: typo — "RandomFields.jl".
- Line 221: Could you add one or two more sentences to elaborate on the significance of the ESS? Why is this an interesting thing to note?
- Line 232: typo — "warning centre's".
- Section 6.2: Considering, say, the 32 ranks per node case, the biggest test runs on only 4 nodes (128 / 32). Yet the parallel efficiency has already dropped to only around 25%. This is a much bigger drop that I would expect, intuitively. Am I missing something? Scaling tests often run into the hundreds of nodes before encountering such limits.
- Figure 6, left: Could you clarify which measure of weak scaling parallel efficiency you employ here? I am guessing it is $E(N) = T(1)/T(N)$, where $T(i)$ is the wall time for running on $i$ processors.
- Line 310: "The standard deviation of the model and observation errors are set to 1 and 10 hPa respectively." — isn't the observation error already stated on line 309? Also what is the "model error" in this case? I thought that the SPEEDY model is integrated without a model error term?

---

## Referee Comment (RC2)

**1 General response**

My overall impression of this work is positive, I think that the authors have done a nice work of outlining a novel data assimilation (DA) framework that is distinguished from a variety of existing ones within the Julia language by focusing on a scalable implementation of particle filtering. I think that this fills a unique gap in the development of DA methodology and shows a lot of potential for research and possibly operational purposes when it is sufficiently mature. However, I do think that there is currently room for improvement in the manuscript including its discussion of the more delicate aspects of particle filtering for DA, and with respect to the framework's numerical validation and proof-of-concept. I outline these points in the major revisions section and I discuss minor items in the section thereafter.

**2 Major Revisions**

1. Section 3.2. State Model Error. There are many important assumptions about the state model error that I think require additional discussion. It is a critical point that the state process model is forced stochastically, as the types of stochastic filters implemented in this iteration of the framework (boot strap and optimal importance sampling) strongly depend on this assumption to maintain sample diversity. Standard multinomial resampling will lead to ensemble collapse if the empirical filtering measure is a weighted sum of Dirac measures when there is no state process randomness to drive the resampled particles apart. However, many geophysical models make no explicit assumption or use of a stochastic paramterization or of a noise processes. Therefore, I feel like this section requires an expanded discussion of the implementation of the stochastic forcing that is compatible with this framework for typical geophysical models. There is an example use-case with the SPEEDY model which is well-appreciated, but a more detailed discussion of the example use-case and a more general discussion of the implementation of random forcing for other state process models would strongly enhance the impact of the manuscript.

2. Section 3.2. Observation Model.

   > "The number of observations and the indices of the observed variables within a multiple dimensional state space model are needed. The locations of the observation stations can be passed within a simple .txt file. The observations can come from an online integration of the state space model or read in from file/sensor."

   This quote leads me to believe that the observation models which are compatible with this framework are assumed to be sparse direct measurements of the state model process. If this is the case, I would like this to be stated directly and to have a discussion of how indirect nonlinear measurements will be integrated into future iterations of the framework, as that would be a major limitation for use in real-world DA. If I have misinterpreted this statement, I would like to have a clearer discussion of how indirect nonlinear measurements are currently integrated into the model-agnostic framework.

3. Localization. The scope of the package aims to be scalable to high-dimensional operational DA. Please give some word about localization in this framework, as this is generically non-trivial to implement, and more so in a model-agnostic framework. However, without including a means for localization at a primal level in this framework, the scalability of the framework may be intrinsically limited, as localization is utilized in virtually every form of currently operational DA as a means to regularize the filtering problem.

4. Numerical Validation, Section 6.1.

   > "As the tsunami model implemented here is linear and Gaussian one can compare the obtained distributions with a ground truth Kalman filter."

   This demonstration should be included here for the reason stated above, and I don't belive this is currently in the manuscript. Please make a direct empirical validation of the particle filter methods versus the optimal Kalman filter in this context for use as a calibration metric and to understand the relationship between sample size and filter performance versus the optimal estimator.

5. Numerical Validation, Section 7.

> "As stated in the introduction, one of the key benefits of particle filters is to provide the promise of non-linear and non-Gaussian DA. To highlight this sample distributions of the surface pressure at various observation locations at different time points are shown in Fig. 8. The distributions across the n = 256 particles exhibit heavy tails towards the true surface pressure at the given locations."

This demonstration of heavy-tails is well-appreciated as this does support the hypothesis that including higher-order information in the estimation of the filtering measure will improve its empirical representation. Nonetheless, the current demonstration I think has missed the opportunity to make a direct validation versus, e.g., a standard flavor of the EnKF as discussed in the surveyed literature. While the current numerical demonstration makes the case that a linear-Gaussian approximation is inaccurate, it doesn't verify that it is inadequate. If the goal is to introduce a framework that addresses intrinsic inadequacies of widely used DA schemes with linear-Gaussian assumptions, a direct demonstration of this point would make a much stronger case for the impact of this framework.

**3   Minor Revisions**

1. Line 25. This references variational DA in the form of 3D- and 4D-VAR, but it makes no citation to related literature. Please expand this with relevant citations and include some references to more modern ensemble-variational methods. See, e.g., Bannister [2017] and refernces therein.

2. Lines 61 - 74. While differing in the intended scope, the literature review does not discuss DataAssimilationBenchmarks.jl and its two key references [Grudzien and Bocquet, 2022, Grudzien et al., 2022]. Please include these in your literature review.

3. Table 1. The table is not currently consistent with the literature review – please include all referenced software packages in Table 1, including EnsembleKalmanProcesses.jl and DataAssimilationBenchmarks.jl.

**References**

R. N. Bannister. A review of operational methods of variational and ensemble-variational data assimilation. *Q. J. R. Meteorol. Soc.*, 143(703):607–633, 2017.

C. Grudzien and M. Bocquet. A fast, single-iteration ensemble Kalman smoother for sequential data assimilation. *Geoscientific Model Development*, 15(20):7641–7681, 2022. doi: 10.5194/gmd-15-7641-2022. URL https://gmd.copernicus.org/articles/15/7641/2022/.

C. Grudzien, C. Merchant, and S. Sandhu. Dataassimilationbenchmarks.jl: a data assimilation research framework. *Journal of Open Source Software*, 7(79):4129, 2022. doi: 10.21105/joss.04129. URL https://doi.org/10.21105/joss.04129.

---

## Author Response (AR1)

**ParticleDA.jl v.1.0: A distributed particle filtering data assimilation package**

Response

July 30, 2023

We would like to thank the reviewers for their helpful comments and feedback; our responses to their main points are outlined below and an updated version of the article with highlighting of the revisions made in response to the comments has been made available.

**1 Reviewer 1**

**1.1 Main comments**

Line 312: My only major comment was triggered by this statement: "It can be seen that the areas of greatest percentage error coincide with areas that lack observation stations." I want to challenge this statement as I don't see such a strong coincidence. Instead the error patterns seem to be dominated by patterns of midlatitude weather systems, which is what you would expect to see if you compared two snapshots of surface pressure in the same model run at different times. In fact, I don't see sufficient evidence that the assimilation is actually working. It probably is, but it's hard to tell unless you compare also with a run without any assimilation. A number of further plots would help to make this section more complete.

Author's response: Please see the updated Figure 8, where we have produced the suggested four sub-figures. We have removed the quoted statement and updated section 7.1.The reviewer is correct in stating that the largest errors in the assimilation run appear to be dominated by mid-latitude patterns. The bottom subplots showcase the benefit of the assimilation. As suggested by the reviewer we have plotted the time averaged error for the mean of the assimilated ensemble and that of an ensemble where no assimilation has occurred. Overall, the assimilated ensemble mean exhibits a lower error.

**1.2 Minor comments**

Our thanks to the reviewer for pointing out the mentioned typos; these have now been corrected.

1. Line 221: Could you add one or two more sentences to elaborate on the significance of the ESS? Why is this an interesting thing to note?

   Author's response: The estimated sample size (ESS) gives an indication of the level of degeneracy in the particle weights; we have added more information about ESS and its usefulness as a metric in Section 2.2.

2. Section 6.2: Considering, say, the 32 ranks per node case, the biggest test runs on only 4 nodes (128 / 32). Yet the parallel efficiency has already dropped to only around 25%. This is a much bigger drop that I would expect, intuitively. Am I missing something? Scaling tests often run into the hundreds of nodes before encountering such limits.

   Author's response: We agree with reviewer that the drop off seen in the previously included parallel efficiency results as the number of ranks increased was unexpectedly quick and we have spent considerable time improving the parallel efficiency of the package. A major factor in the previous loss in performance was found to be load imbalance across the threads. The problem has been partially remedied by using a dynamic task based thread allocation instead of a static thread model. New scaling results are now included where the model is run on up to 16 nodes on ARCHER2. The new results showcase near optimal weak scaling efficiency for single node cases but with a drop off in performance when going to multi-node runs. Therefore, future work for the package will focus on an improved load imbalancing across nodes.

3. Figure 6, left: Could you clarify which measure of weak scaling parallel efficiency you employ here? I am guessing it is $E(N) = T(1)/T(N)$, where $T(i)$ is the wall time for running on $i$ processors.

**Author's response:** We are using the following defined weak scaling efficiency: $E(N) = T(2)/T(N)$. This has been clarified in Section 6.2.

4. Line 310: "The standard deviation of the model and observation errors are set to 1 and 10 hPa respectively." — isn't the observation error already stated on line 309? Also what is the "model error" in this case? I thought that the SPEEDY model is integrated without a model error term?

**Author's response:** In order to clarify the model setup for the SPEEDY case we have now provided additional information in Section 7.

**2 Reviewer 2**

**2.1 Main comments**

1. Section 3.2. State Model Error. There are many important assumptions about the state model error that I think require additional discussion. It is a critical point that the state process model is forced stochastically, as the types of stochastic filters implemented in this iteration of the framework (bootstrap and optimal importance sampling) strongly depend on this assumption to maintain sample diversity. Standard multinomial resampling will lead to ensemble collapse if the empirical filtering measure is a weighted sum of Dirac measures when there is no state process randomness to drive the resampled particles apart. However, many geophysical models make no explicit assumption or use of a stochastic parameterization or of a noise processes. Therefore, I feel like this section requires an expanded discussion of the implementation of the stochastic forcing that is compatible with this framework for typical geophysical models. There is an example use-case with the SPEEDY model which is well-appreciated, but a more detailed discussion of the example use-case and a more general discussion of the implementation of random forcing for other state process models would strongly enhance the impact of the manuscript.

**Author's response:** We agree with the reviewer of the importance of the assumption of the state process model being stochastic for the proposal distributions used here to maintain diversity in the ensemble, and have added text to emphasise this point in Section 2.1. We have also added a paragraph discussing the requirement for the state to evolve stochastically over time when introducing the state space model formulation assumed in Section 2, and noted this may not the case for the usual formulation of geophysical models. For each of the example models considered which are based on solving a system of ordinary or partial differential equations (Sections 5, 6 and 7) we have also included an explicit description of how stochasticity is introduced in the state dynamics.

2. Section 3.2. Observation Model.
   *"The number of observations and the indices of the observed variables within a multiple dimensional state space model are needed. The locations of the observation stations can be passed within a simple .txt file. The observations can come from an online integration of the state space model or read in from file/sensor."*
   This quote leads me to believe that the observation models which are compatible with this framework are assumed to be sparse direct measurements of the state model process. If this is the case, I would like this to be stated directly and to have a discussion of how indirect nonlinear measurements will be integrated into future iterations of the framework, as that would be a major limitation for use in real-world DA. If I have misinterpreted this statement, I would like to have a clearer discussion of how indirect nonlinear measurements are currently integrated into the model-agnostic framework.

**Author's response:** We thank the reviewer for alerting us to this inconsistency in the text; the quoted text was originally written for an earlier version of the package which did indeed assume sparse direct measurements of the state model process at a set of point locations. The package now allows for more general observation models, with filtering using the bootstrap proposal able to be applied to any observation model compatible with Eq. 1 (that is for which the observations at a particular time index depend only on the current state and for which the conditional distribution given the state has a tractable density function), which would allow for example non-linear observation operators. For a filter using the locally optimal proposal, a more restricted linear-Gaussian observation model (Eq. (3) in the paper) is assumed to ensure the local optimal proposal distribution is analytically tractable. While in this case the observations are assumed to depend linearly on the state, they do not necessarily need to correspond to direct (noisy) measurements. We have clarified these points in the text and removed the previous inconsistent description.

3. Localization. The scope of the package aims to be scalable to high-dimensional operational DA. Please give some word about localization in this framework, as this is generically non-trivial to implement, and more so in a model-agnostic framework. However, without including a means for localization at a primal level in this framework, the scalability of the framework may be intrinsically limited, as localization is utilized in virtually every form of currently operational DA as a means to regularize the filtering problem.

Author's response: The authors agree with the reviewer of the importance of spatial localization in scaling particle filter based data assimilation to high-dimensional state space models, specifically spatially extended models with densely distributed point observations, and adding support for this to ParticleDA.jl is one of our priorities for future development of the package, as we now mention in both the introduction and conclusion sections of the paper. With regards to implementing this in a model-agnostic fashion, while we agree this is non-trivial, we believe this should be feasible to do using a similar approach to that currently implemented (and described in Section 3.2 in the revised paper) for the locally optimal proposal for state space models with linear-Gaussian substructure. Specifically, the base interface required to be implemented by models can be extended with functions exposing additional structure that we wish to exploit in a particular class of filters. Specifically for spatially localized particle filters we would need functions to evaluate (i) the spatial distance between the observation point associated with an index of the observation vector and the spatial mesh node associated with an index of the state vector and (ii) the spatial distance between the spatial mesh nodes associated pairs of indices in the state vector. One of the authors has previously worked on implementing spatially-localised particle filtering methods in a model-agnostic data assimilation framework in Python (`https://github.com/thiery-lab/data-assimilation/`) and we plan to use the design there to inform the implementation in ParticleDA.jl.

4. Numerical Validation, Section 6.1.
"As the tsunami model implemented here is linear and Gaussian one can compare the obtained distributions with a ground truth Kalman filter."
This demonstration should be included here for the reason stated above, and I don't believe this is currently in the manuscript. Please make a direct empirical validation of the particle filter methods versus the optimal Kalman filter in this context for use as a calibration metric and to understand the relationship between sample size and filter performance versus the optimal estimator.

Author's response: This has indeed already been carried out, with the results plotted in Figure 6 which shows the error in filtering estimates of the mean at each observation time *as compared to the true mean computed using a Kalman filter*. Further clarifying information has been added to Section 6.1 and the axis labels in Figure 6 updated to make this clearer.

5. Numerical Validation, Section 7.
"As stated in the introduction, one of the key benefits of particle filters is to provide the promise of non-linear and non-Gaussian DA. To highlight this sample distributions of the surface pressure at various observation locations at different time points are shown in Fig. 8. The distributions across the $n = 256$ particles exhibit heavy tails towards the true surface pressure at the given locations."
This demonstration of heavy-tails is well-appreciated as this does support the hypothesis that including higher-order information in the estimation of the filtering measure will improve its empirical representation. Nonetheless, the current demonstration I think has missed the opportunity to make a direct validation versus, e.g., a standard flavor of the EnKF as discussed in the surveyed literature. While the current numerical demonstration makes the case that a linear-Gaussian approximation is inaccurate, it doesn't verify that it is inadequate. If the goal is to introduce a framework that addresses intrinsic inadequacies of widely used DA schemes with linear-Gaussian assumptions, a direct demonstration of this point would make a much stronger case for the impact of this framework.

Author's response: We agree that this is an important point. However, we deem it to be beyond the scope of the work outlined here. We have made explicit reference to the work of (Miyoshi et al., 2014 and Konda and Miyoshi (2019)) in Section 7.1, where a very similar experimental set-up was used with an ensemble Kalman filter to produce non-Gaussian distributions. However, a key difference are the relative ensemble sizes, with 256 particles used in our case to illustrate the heavy tails versus an ensemble of 10,240 members used in Miyoshi et al., 2014 and Konda and Miyoshi (2019).

**2.2 Minor Comments**

1. Line 25. This references variational DA in the form of 3D- and 4D-VAR, but it makes no citation to related literature. Please expand this with relevant citations and include some references to more modern ensemble-variational methods. See, e.g., Bannister [2017] and references therein.

   Author's response: Additional references have been added.

2. Lines 61 - 74. While differing in the intended scope, the literature review does not discuss DataAssimilationBenchmarks.jl and its two key references [Grudzien and Bocquet, 2022, Grudzien et al., 2022]. Please include these in your literature review.

   Author's response: The additional references have been added.

3. Table 1. The table is not currently consistent with the literature review – please include all referenced software packages in Table 1, including EnsembleKalmanProcesses.jl and DataAssimilationBenchmarks.jl.

   Author's response: The table is a summary of the literature review with only Julia packages which are similar in design and purpose to ParticleDA.jl included.

---

## Referee Report (RR1)

I appreciate the authors' efforts to build a parallelizable model-agnostic particle filter system, which holds promise for various geoscience applications. Nevertheless, I would like to raise some concerns about the state transition density, or the model error Q, in the following:

1. The difference of the particle weights between using the bootstrap and using the optimal proposal is determined by Q (e.g., see the reference below). Therefore, the way that Q is specified determines how much the optimal proposal outperforms the bootstrap proposal. This could be added into the discussion to emphasize the importance of the choice of Q.

   *e.g., see Sections 9.2.2-9.2.3 in*
   *Evensen, Geir, Femke C. Vossepoel, and Peter Jan van Leeuwen. Data assimilation fundamentals: A unified formulation of the state and parameter estimation problem. Springer Nature, 2022.*

2. The particle filter algorithm itself is model-agnostic, while Q is not. Although this manuscript has provided examples illustrating the generation of Q, in general this is not trivial for many geophysical models. For example, not only do we need to consider the smoothness of the state variable, but also the physical constraints across different variable types. For example, the wind and pressure should largely satisfy the geostrophic balance relation in the AGCM. In addition, for many geoscience applications, Q can be state dependent. For example, the model errors for predicting a heat wave can be very different from predicting a hurricane. Probably beyond the scope of this study, the transition density can also be quite non-Gaussian, e.g., for modeling the convection process in weather prediction models.

   The user might need to build Q for their own model, which again, is not trivial for many geophysical models. Since this manuscript is submitted to GMD, I would recommend expand the discussions surrounding the construction of Q for geoscience applications (e.g., in the conclusion section).

3. I would recommend elaborate more on how *get_covariance_state_noise* and *get_covariance_state_observation_given_previous_state* are being evaluated. Are there any computational challenges to evaluate these two functions for a very high-dimensional (e.g., $d_x \sim 10^9$) model?

I have a few other minor comments:

1. For a spatially extended model, like a weather prediction model, the dimension can be as high as $10^9$. Will this be an issue when copying a state from one processor to another? How does the overall algorithm scale with the dimension of the model state $d_x$?

2. I find it somewhat less convincing that the particle filter can work better than any existing linear and Gaussian DA methods in the experiments (e.g., from the results in Figs 8-9). Nevertheless, I do understand the primary goal of this manuscript is to showcase the capabilities of the package, instead of proposing a novel particle filter methodology and conduct comprehensive comparisons with existing methods, etc.

   Therefore, I do not insist but recommend, e.g., add a new experiment with non-linear observation operator, and/or compare the performance of PF and an ensemble Kalman filter against the ground truth in an OSSE (e.g., in Figure 8, you could also add a panel that shows the results from using an ensemble Kalman filter).

3. Figure 8 -> is the unit of the time averaged error correct? An error in surface pressure exceeding 50 hPa seems unrealistically large.

---

## Referee Report (RR2)

I appreciate the authors' responses in answering my questions in the previous iteration. The revised manuscript addresses most of my concerns. In my opinion, the manuscript is nearly ready for publication, with a few minor issues below that need addressing.

Line 336: I am curious about why the locally optimal proposal cannot be applied for the non-linear observation operator? Even if the linear assumption is invalid, shouldn't it still be applicable? (like we can still apply the ensemble Kalman filter to assimilate non-linear observation even if it's not optimal) Please clarify if there's anything that I might be missing here.

Line 344: "The time averaged RMSE results are plotted in 5." -> Figure 5.

Line 344-345: "The effect of the non-linear observation operator can be clearly seen, where the time averaged RMSE for the linear case is lower in all set-ups".

This statement is inaccurate. There are a lot of factors determining the magnitude of RMSE. Assimilating a linear observation (i.e., with linear observation operator) with very large observation error standard deviation can also lead to a large RMSE. When the observation error standard deviation is fixed, using different observation operator leads to different shape of likelihood function, and therefore different magnitude of RMSE. I recommend rephrasing this sentence or just removing it.

Line 449: what are the units for the standard deviation (especially for ps)?

Line 459: "10 hPa" noises in ps seems to be quite large

Figure 9: I am still skeptical about the RMSE values in these experiments. Even in a model run without DA (the lower left panel), the RMSE are unrealistically large, suggesting the large RMSE is not related to the observing system. I suspect that this is a result of the experiment setup that the ps standard deviation in Q is set to 100 (hPa) and the observation error is set to 10 (hPa). I think setting the standard deviation as 100 (Pa) = 1 (hPa) and 10 (Pa) = 0.1 (hPa) could lead to a more realistic representation.

---

## Author Response (AR2)

**ParticleDA.jl v.1.0: A distributed particle filtering data assimilation package**

Response 2

November 16, 2023

We would like to thank the reviewers and editor for their helpful comments and feedback. Our response to their points are outlined below. An updated version of the article with highlights of the changes made has been uploaded.

**1 Topic Editor**

This manuscript is an excellent contribution to the DA and geoscience communities. As all reviewers think highly of this manuscript, I recommend that the manuscript needs significant revision to address reviewer 3's comments and suggestions. In particular, I agree with reviewer 3's comments that the authors should add a new experiment with nonlinear observation operator to emphasize the robustness of the package in dealing with nonlinear dynamics, which geophysical models commonly deal with. Also, the Lorenz model is a very common simple model for evaluating innovative DA algorithms. Therefore, the authors should follow the setup of the popular references and provide the analysis RMSE to give some basic idea about the performance of the new ParticleDA.jl v.1.0 package. Currently, the choice of the observation error variance is very small.

Author's response: Taking the suggestions on board section 5.1 has been added which showcases the performance of the bootstrap filter with a non-linear observation operator. Further, the performance of the locally optimal proposal for varying numbers of particles on the Lorenz 63 system is now highlighted in the updated Fig. 4, please be advised that the observation error variance has been increased in this case.

**2 Reviewer 3**

I appreciate the authors' efforts to build a parallelizable model-agnostic particle filter system, which holds promise for various geoscience applications. Nevertheless, I would like to raise some concerns about the state transition density, or the model error Q, in the following:

1. The difference of the particle weights between using the bootstrap and using the optimal proposal is determined by Q (e.g., see the reference below). Therefore, the way that Q is specified determines how much the optimal proposal emphasize the importance of the choice of Q.

   e.g., see Sections 9.2.2-9.2.3 in Evensen, Geir, Femke C. Vossepoel, and Peter Jan van Leeuwen. Data assimilation fundamentals: A unified formulation of the state and parameter estimation problem. Springer Nature, 2022.

   The particle filter algorithm itself is model-agnostic, while Q is not. Although this manuscript has provided examples illustrating the generation of Q, in general this is not trivial for many geophysical models. For example, not only do we need to consider the smoothness of the state variable, but also the physical constraints across different variable types. For example, the wind and pressure should largely satisfy the geostrophic balance relation in the AGCM. In addition, for many geoscience applications, Q can be state dependent. For example, the model errors for predicting a heat wave can be very different from predicting a hurricane. Probably beyond the scope of this study, the transition density can also be quite non-Gaussian, e.g., for modeling the convection process in weather prediction models.

   The user might need to build Q for their own model, which again, is not trivial for many geophysical models. Since this manuscript is submitted to GMD, I would recommend expand the

discussions surrounding the construction of Q for geoscience applications (e.g., in the conclusion section).

*Author's response:* The author's recognise that this is a key ingredient and was a focus in the previous round of revision. Please see lines 110 - 112 where the importance of maintaining physical constraints and relationships of the underlying physics is highlighted. To further expand on this point we have added a paragraph in conclusion section on this, please see lines 502 - 509.

2.  I would recommend elaborate more on how *get_covariance_state_noise* and *get_covariance_state_observation_given_previous_state* are being evaluated. Are there any computational challenges to evaluate these two functions for a very high-dimensional (e.g., $d_x \sim 10^9$) model?

*Author's response:* The authors agree that this is a key point, and we have added further detail to manuscript in Section 3.2 (*Model interface*) describing how the implementation allows exploiting sparsity in the observation operator $H$ to keep the computational and memory costs manageable when working with models with high-dimensional state spaces.

**2.1 Minor comments**

1.  For a spatially extended model, like a weather prediction model, the dimension can be as high as $10^9$. Will this be an issue when copying a state from one processor to another? How does the overall algorithm scale with the dimension of the model state $d_x$?

*Author's response:* This is a very important point with the key question focusing on how the package will perform when the spatial model is also built upon a distributed memory set up. For example weather prediction models traditionally rely on MPI (Message Passing Interface) to run across multiple compute nodes and how ParticleDA.jl performs in this case is the focus of ongoing development but is currently beyond the scope of this work.

2.  I find it somewhat less convincing that the particle filter can work better than any existing linear and Gaussian DA methods in the experiments (e.g., from the results in Figs 8-9). Nevertheless, I do understand the primary goal of this manuscript is to showcase the capabilities of the package, instead of proposing a novel particle filter methodology and conduct comprehensive comparisons with existing methods, etc.
    Therefore, I do not insist but recommend, e.g., add a new experiment with non- linear observation operator, and/or compare the performance of PF and an ensemble Kalman filter against the ground truth in an OSSE (e.g., in Figure 8, you could also add a panel that shows the results from using an ensemble Kalman filter).

*Author's response:* As suggested an additional experiment (section 5.1) has been added with a non-linear observation operator for the Lorenz 63 system. The introduction of this non-linear operator influences the performance of the bootstrap proposal. We also agree that a comparison of the particle filter against an ensemble Kalman filter for the AGCM case is beyond the scope here, but references have been made to the performance of a Local Ensemble Kalman Transform Filter (LETKF) in a similar non-linear set-up (see lines 473-476).

3.  Figure 8: is the unit of the time averaged error correct? An error in surface pressure exceeding 50 hPa seems unrealistically large.

*Author's response:* The units on the time averaged error are correct. The authors are aware that this is quite large in comparison to other studies but acknowledge that in this set up only the surface pressure is observed and on a very sparse observation network. The purpose of Fig. 9 (previously Fig. 8) is to emphasis the performance of the particle filter against an ensemble where no assimilation is occurring and to showcase the interoperability of the package with a pre-exisiting Fortran codebase. Please see the discussion in the previous round of revisions.

---

## Author Response (AR3)

**ParticleDA.jl v.1.0: A distributed particle filtering data assimilation package**

**Response 3**

**January 13, 2024**

We would like to thank the reviewers and editor for their helpful comments and feedback. Our response to their points are outlined below. An updated version of the article with highlights of the changes made has been uploaded.

**1 Reviewer 3**

I appreciate the authors' responses in answering my questions in the previous iteration. The revised manuscript addresses most of my concerns. In my opinion, the manuscript is nearly ready for publication, with a few minor issues below that need addressing.

1. Line 336: I am curious about why the locally optimal proposal cannot be applied for the non-linear observation operator? Even if the linear assumption is invalid, shouldn't it still be applicable? (like we can still apply the ensemble Kalman filter to assimilate non-linear observation even if it's not optimal) Please clarify if there's anything that I might be missing here.

   Author's response: This is an interesting question, and in part depends what is specifically meant by the *locally optimal proposal*.

   If we take the *locally optimal proposal* to specifically refer to the distribution with density defined by equation (8) in the paper, then assuming a non-linear observation model (but continuing to assume Gaussian observation noise with a fixed variance), that is $\boldsymbol{y}_t \mid \boldsymbol{x}_t \sim \mathcal{N}(h_t(\boldsymbol{x}_t), R)$, an immediate issue in evaluating the density in equation (8) or sampling from the corresponding distribution is that we now have a nonlinear operator $h_t : \mathbb{R}^{d_x} \to \mathbb{R}^{d_y}$ rather than a linear operator represented by a matrix $H \in \mathbb{R}^{d_x \times d_y}$. One obvious approach to evaluating the matrix products involving $H$ in equation (8) would be to use a linearization of $h_t$ around the input as a substitute for $H$, that is $H \approx \partial h_t(\boldsymbol{x}_t)$, the Jacobian of $h_t$ evaluated at $\boldsymbol{x}_t$. If we adopted this approach then we could evaluate the density in equation (8) and sample from the corresponding distribution, however, the resulting proposal distribution would no longer be the locally optimal (in the sense of minimising the variance of the importance weights) nor would the expression for the importance weights in equation (9) be valid (as terms in the density ratio which exactly cancel in the linear case would no longer do so in the non-linear approximation). We could use the underlying definition of the importance weights as a density ratio given in the equation in step 5 of Algorithm 1 to compute valid importance weights, but importantly these would depend on both $\boldsymbol{x}_t$ and $\boldsymbol{x}_{t-1}$ (sampled state proposal and previous state) unlike the true locally optimal proposal, for which by construction the importance weights depend only on the previous state (and so have no variance contribution from the proposal distribution). So we can use a proposal which is an approximation to the locally optimal proposal for the linear-Gaussian observation case, but the resulting proposal is not the locally optimal proposal for the model.

   If we instead consider the general definition of the density of the locally optimal proposal in equation (6) and corresponding expression for the importance weights in equation (7), then for an observation model $\boldsymbol{y}_t \mid \boldsymbol{x}_t \sim \mathcal{N}(h_t(\boldsymbol{x}_t), R)$, the integral appearing in both equations (6) and (7) is analytically intractable. We could potentially estimate this integral and use this to form an approximation to the locally optimal proposal distribution, with in comparison to the approach above we at least in this situation directly approximating the distribution of interest, rather than a proxy to that distribution. Depending on exactly on how we estimate the integral though, the resulting proposal distribution may no longer be from a known parametric family and so it may be challenging to generate independent samples from.

2. Line 344: "The time averaged RMSE results are plotted in 5." -¿ Figure 5.

   Author's response: The missing Fig. has been added.

3. Line 344-345: "The effect of the non-linear observation operator can be clearly seen, where the time averaged RMSE for the linear case is lower in all set-ups".

   This statement is inaccurate. There are a lot of factors determining the magnitude of RMSE. Assimilating a linear observation (i.e., with linear observation operator) with very large observation error standard deviation can also lead to a large RMSE. When the observation error standard deviation is fixed, using different observation operator leads to different shape of likelihood function, and therefore different magnitude of RMSE. I recommend rephrasing this sentence or just removing it.

   Author's response: Thank you for pointing this out, this statement has now been dropped.

4. Line 449: what are the units for the standard deviation (especially for ps)?

   Author's response: The units have been added.

5. Line 459: "10 hPa" noises in ps seems to be quite large

   Author's response: Please see the response below.

6. Figure 9: I am still skeptical about the RMSE values in these experiments. Even in a model run without DA (the lower left panel), the RMSE are unrealistically large, suggesting the large RMSE is not related to the observing system. I suspect that this is a result of the experiment setup that the ps standard deviation in Q is set to 100 (hPa) and the observation error is set to 10 (hPa). I think setting the standard deviation as 100 (Pa) = 1 (hPa) and 10 (Pa) = 0.1 (hPa) could lead to a more realistic representation.

   Author's response: To clarify the existing experiment was run with the ps standard deviation in Q set to 100 Pa and the standard deviation of the additive observation noise set to 1000 Pa. The authors apologise for the confusion and have added the units to the appropriate section. The authors would also like to further highlight a mistake made in the previous round. As stated by the reviewer the $L_2$ error is indeed too large. Unfortunately, the appropriate scaling of the pressure fields was not carried out correctly when producing the figures. The corrected figure (Fig. 1) is produced below. We would like to thank the reviewer for pointing this out and apologise for the oversight.

   To ensure that we have fully engaged with the reviewer's feedback Fig. 2 is the output of the suggested set up, where the observation error standard deviation is set to 10 Pa. Differences between Fig. 1 and Fig. 2 can be seen but when assessing the estimated ensemble size for this run (not shown here) we see occurrences of particle degeneracy. Therefore the original setup with the updated figure (Fig. 1) has been retained in the manuscript.

[Figure]

Figure 1: Corrected Fig 9. from the manuscript.

[Figure]

Figure 2: Reviewer's suggested setup with the standard deviation of the additive observation noise set to 10 Pa.